# Differential ion mobility mass spectrometry in immunopeptidomics identifies neoantigens carrying colorectal cancer driver mutations

Yuriko Minegishi[1], Kazuma Kiyotani [2], Kensaku Nemoto[2], Yoshikage Inoue[3], Yoshimi Haga [1], Risa Fujii[1], Naomi Saichi[1], Satoshi Nagayama[4] & Koji Ueda [1✉]

Understanding the properties of human leukocyte antigen (HLA) peptides (immunopeptides) is essential for precision cancer medicine, while the direct identification of immunopeptides from small biopsies of clinical tissues by mass spectrometry (MS) is still confronted with technical challenges. Here, to overcome these hindrances, high-field asymmetric waveform ion mobility spectrometry (FAIMS) is introduced to conduct differential ion mobility (DIM)-MS by seamless gas-phase fractionation optimal for scarce samples. By established DIM-MS for immunopeptidomics analysis, on average, 42.9 mg of normal and tumor colorectal tissues from identical patients (n = 17) were analyzed, and on average 4921 immunopeptides were identified. Among these 44,815 unique immunopeptides, two neoantigens, KRAS-G12V and CPPED1-R228Q, were identified. These neoantigens were confirmed by synthetic peptides through targeted MS in parallel reaction monitoring (PRM) mode. Comparison of the tissue-based personal immunopeptidome revealed tumor-specific processing of immunopeptides. Since the direct identification of neoantigens from tumor tissues suggested that more potential neoantigens have yet to be identified, we screened cell lines with known oncogenic KRAS mutations and identified 2 more neoantigens that carry KRAS-G12V. These results indicated that the established FAIMS-assisted DIM-MS is effective in the identification of immunopeptides and potential recurrent neoantigens directly from scarce samples such as clinical tissues.

[1] Cancer Proteomics Group, Cancer Precision Medicine Center, Japanese Foundation for Cancer Research, Tokyo, Japan. [2] Project for Immunogenomics, Cancer Precision Medicine Center, Japanese Foundation for Cancer Research, Tokyo, Japan. [3] Department of Surgery, Kyoto University, Kyoto, Japan. [4] Development of Gastroenterological Surgery, Cancer Institute Hospital of Japanese Foundation for Cancer Research, Tokyo, Japan. ✉email: koji.ueda@jfcr.or.jp

Human leukocyte antigen (HLA) class I is the major histocompatibility complex (MHC) of humans and serves roles in self-tolerance and innate immunity. The HLA complex contains an HLA peptide (HLAp, also called immunopeptide) that plays a role in self-nonself discrimination. In addition to the oncogenic virus proteins, mutated proteins or the epigenetically significantly increased normal proteins in cancer cells could be a source[1] of so-called neoantigens that fill critical roles in cancer immunity. With the rapid progress in cancer immunotherapy such as the new generation of TCR-like CARs, TCR-CARs[2], and the bispecific antibodies[3], the information about shared neoantigens across cancer patients to benefit by such these therapies is highly required. The neoantigen that carries well-known oncogenic mutation is especially an attractive target for this purpose. The neoantigens presented on the surface of cancer cells are recognized as nonself antigens by the relevant repertoire of T cells and can trigger anticancer immunity[4]. Immune checkpoint inhibitors (ICIs) have been established to invigorate this anticancer immune system as cancer immunotherapies[5]. Although immunotherapies are efficacious against certain cancer types[5], there is still a lack of knowledge about predictive factors for successful ICI indication[5,6]. From these perspectives, the comparative analysis of normal and tumor tissue-based immunopeptidomes from identical patients is increasingly important to avoid the adverse events of ICI in patients[7].

Mass spectrometry (MS) immunopeptidomics is currently the sole method that can directly identify the immunopeptides presented on the cellular surface. According to the guidelines for immunopeptidomics[8,9], one of the current disadvantages of MS-based immunopeptidomics is the inability to analyze a scarce sample, such as the tissue of an endoscopic biopsy, compared to the genomic prediction of immunopeptide in silico[10].

In the previous method for immunopeptidomics, to obtain thousands of convincing immunopeptides, more than $1e^8$ cultured cells or nearly 1 g of tissue samples were required[11–13]. In addition, to ensure the best results, the process of chemical pre-fractionation is inevitable, which will become an obstacle to the analysis of scarce samples. In fact, the detection of neoantigens directly from solid tumor tissues (such as colorectal, liver, or ovarian cancer) in previous cases has been unsuccessful[9,14,15]. In the successful case of melanoma detection, at least 0.1 mg of tissue sample with at least 5 mL of lysis buffer was used for sample preparation[11]. While we were submitting this manuscript, the successful identification of neoantigens directly from 1.5 g of colorectal tumor tissue was reported[16]. Using a relatively large mass of tumor tissues surmounted the incapability of unsuccessful identification of neoantigens directly from tissue samples, while it has not yet achieved a fundamental solution. Tumor size varies widely in patients and across types of cancers; thus, it is not always possible to secure a large amount of tumor mass. Alternatively, patient-derived cancer cell line materials, organoids, and xenograft models were often used to secure the required amount of materials in the previous studies[17–19]. This poses the risk of missing knowledge of immunopeptides at the tissue level due to the lack of a cancer-associated microenvironment that affects immunopeptide processing and presentation. Therefore, understanding the cancer immunopeptidome at the tissue level, especially with the appropriate normal control from the same individual, is essential. Furthermore, neoantigens derived from known oncogenic mutations are attractive targets as shared neoantigens for rapidly progressing cancer immunotherapies. From this perspective, establishing a more efficient approach that enables in-depth immunopeptidomics analysis from scarce tissues is indispensable in the field of advanced precision cancer immunotherapy.

It has been written in the meeting report of the Human Immuno-Peptidome Project Consortium (HIPP) that "inability to analyze immunopeptidomes from small amounts of biological material" is one of the main challenges in immunopeptidomics[8]. We hypothesized that this hindrance is, at least in part, due to the inevitable loss of immunopeptides during the general chemical fractionation using columns. To overcome that limitation, we adopted differential ion mobility (DIM)-MS equipped with the interface of high-field symmetric waveform ion mobility mass spectrometry (FAIMS) to eliminate the possibility of unexpected sample loss, which is suitable for the analysis of scarce samples. The seamless gas phase fractionation by the FAIMS interface ensured a sufficiently deep immunopeptidomics analysis so that neoantigens became detectable directly from ~40 mg of colorectal cancer (CRC) tissue samples. Since one of the neoantigens identified from CRC tissue carried oncogenic KRAS (G12V), a major cancer driver mutation, we further explored neoantigens with oncogenic KRAS mutations in known colon cancer cell lines. As a result, two neoantigens carrying KRAS-G12V were identified. Here, we show how established DIM-MS immunopeptidomics analysis can help provide insights into the cancer-specific processing of immunopeptides and the presentation of potential neoantigens.

## Results

### Efficient identification of immunopeptide from scarce samples by established global-immunopeptidomics analysis.
For the sample preparation of Class I immunopeptide in our study, we used W6/32 antibody for immunopurification of HLA complex (See Supplementary Methods). First, we fully optimized the parameters for DIM-MS to maximize the identification of class I immunopeptides by the collection of three fractions at a time. In particular, a set of 9 compensation voltages (CVs), from −80 to −35 V, were selected based on the preferred distribution of immunopeptides confirmed by pilot analyses (Supplementary Fig. 1a). We then further assessed the combination of CVs for the best identification efficiency (Supplementary Fig. 1b). The best performance of gas phase fractionation of immunopeptidomics analysis was established as three CVs per run and three sets of distinct CVs per sample (Fig. 1a). Other conditions, such as collision mode, detector type, maximum injection time, and scan speed, were also fitted accordingly (see Methods). Then, we verified whether the neoantigens could be identified by a database search against the FASTA file containing only somatic mutations of single amino acid substitutions. From the validation studies with or without FAIMS conditions for HCT116 cells shown in Supplementary Fig. 2a, FAIMS-assisted immunopeptidomics analysis using $5e^6$ HCT116 cells identified an average of 3366 immunopeptides, resulting in a 1.2-fold increase over the condition without FAIMS (average of 2772 immunopeptides, Fig. 1b, Supplementary Data 1a and 1b). According to the increase in immunopeptides, the number of identified source proteins was also greater in the FAIMS-assisted condition (2381 proteins on average) than in the without-FAIMS condition (2077 proteins on average, Fig. 1c). This result was consistent throughout the experiments and became statistically significant when the sample scale was increased to $1e^7$ cells (average 4126 identifications with FAIMS) or $1.5e^7$ cells (average 4348 immunopeptides with FAIMS, Fig. 1b). By increasing the sample size under FAIMS conditions, the frequency of neoantigen detection was correspondingly increased (Fig. 1d). Furthermore, FAIMS not only increased the number of identified peptides but also increased the purity of the identified peptides, i.e., the ratio of non-binders by NetMHCpan prediction against the peptide groups or the deduplicated candidates was decreased under FAIMS-assisted

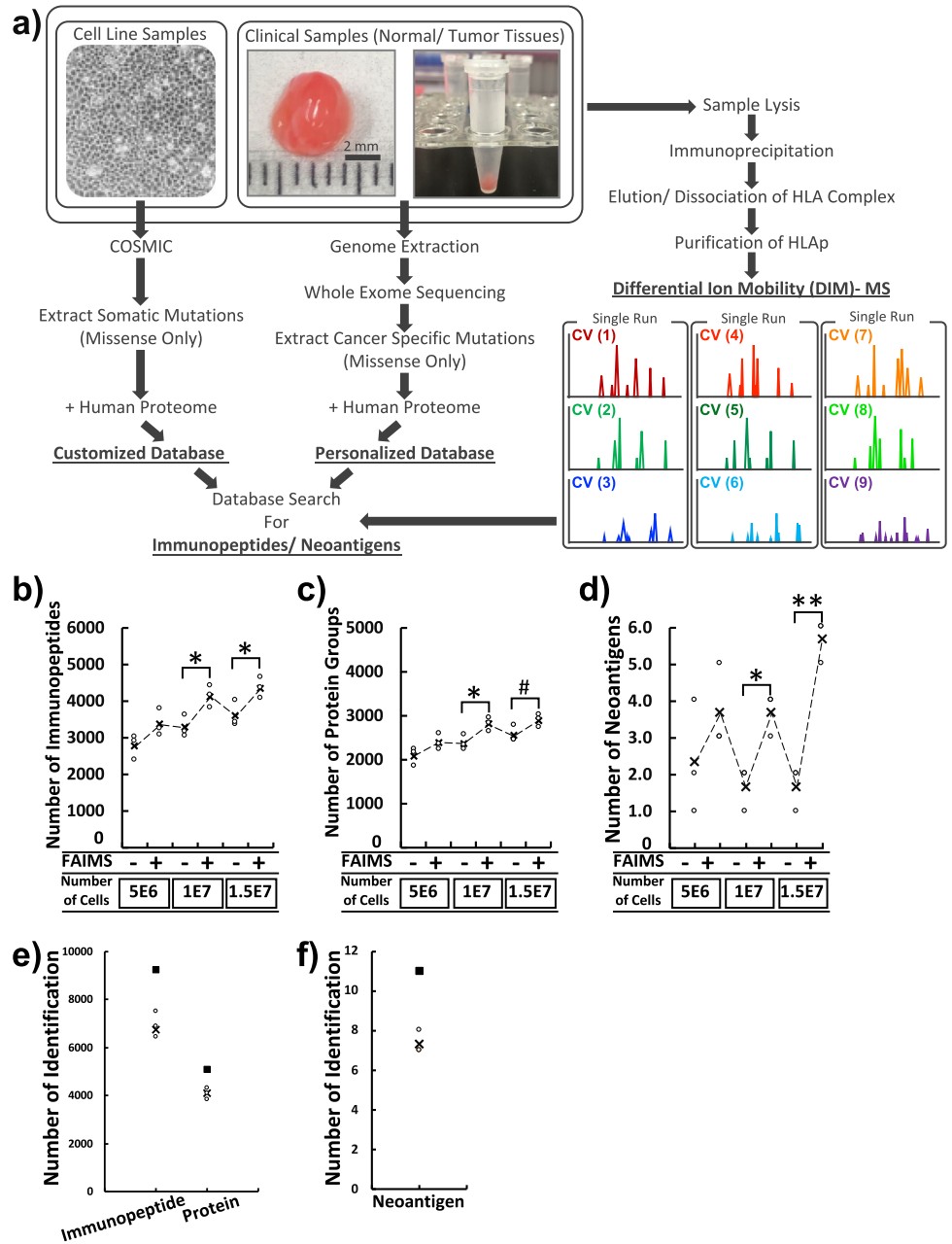

**Fig. 1 A schematic workflow and validation of global-immunopeptidomics analysis by FAIMS-assisted DIM-MS. a** The workflow for proteogenomic personalized immunopeptidomics analysis is shown. The DIM-MS analysis was performed with seamless gas-phase fractionations by installing the FAIMS-Pro interface. Acquired raw files were searched against the tailored database that includes the normal human proteome with sample-corresponding cancer-specific somatic mutations. CV compensation voltage. **b–d** The dot-plots depict the number of identification of immunopeptides, source proteins, and neoantigens in HCT116 cells respectively. The open circles indicate individual data points of three replicates. X indicates the average number of identification from the replicates. Filled box indicates total number of identification from the replicates. The dashed line represents the transition of average number of identification. # $p < 0.1$, * $p < 0.05$, ** $p < 0.01$. **e, f** The dot-plots depict the number of immunopeptides, source proteins, and neoantigens identified in HCT116 cells by global-immunopeptidomics analysis. The number of immunopeptides identified from three independent analyses using 1e8 HCT116 cells. The open circles indicate individual data points of three independent analyses. X indicates the average number of identification from the analyses. Filled box indicates total number of identification from the analyses.

conditions (Supplementary Fig. 2b and c). These results indicated that gas-phase fractionation applied by DIM-MS, with an order of magnitude fewer cells or mass, exhibited the same or better analytical ability for immunopeptidomics than any previously reported methodology[11,12,14]. From three independent analyses using 1e8 HCT116 cells, an average of 6915 immunopeptides from 4109 source proteins were identified, and a total of 9249 unique immunopeptides were derived from 5103 unique source

proteins (Fig. 1e, Supplementary Data 1c). The representative results of immunopeptidomics analysis of HCT116 cells with respect to the CV distribution and the overlap are shown in Supplementary Fig. 3. The typical length (9–15 amino acids) of class I immunopeptides and the reasonable assignment of identified immunopeptides for the corresponding HLA allotype of HCT116 were observed within the immunopeptidome (Supplementary Fig. 4). From this immunopeptidome, an average of 7.3

neoantigens were identified in each independent experiment, resulting in a total of 11 neoantigens (Fig. 1f). As representative neoantigens, 3 out of 11 neoantigens (GAPDH-I69T [p.62-70], CHMP7-A324T [p.316-325], and NAPA-A181V [p.180-188]) identified were further verified by corresponding synthetic peptides under the same method of DIM-MS (Supplementary Fig. 5). For more objective validation, we used the targeted-MS method to obtain the dot product (dotp) score by *Skyline* software. The MS2 spectrum obtained from HCT116 cells and the corresponding synthetic peptides exhibited convincing dotp scores (dotp > 0.95, Supplementary Fig. 6). Seven out of 11 neoantigens identified were not found in the immune epitope database (IEDB) (http://www.iedb.org/home_v3.php, as of 27 November 2020) (Table 1). These results indicated that the FAIMS-applied DIM-MS system could provide an ideal technological platform for in-depth profiling of immunopeptides, as well as direct detection of neoantigens.

**Global-immunopeptidomics enabled direct identification of neoantigens from scarce tissues of CRC.** By established global-immunopeptidomics, CRC samples of tumor or adjacent normal regions of CRC tissues from the same individuals ($n = 17$) were subjected to personalized immunopeptidomics analysis. The general clinical information of the samples, including pathological classification, is listed in Supplementary Data 2a. The mutation load, oncogenic KRAS status, and genetic background of class I HLA are listed in Supplementary Data 2b. Although there was no statistically significant difference in tissue weight used ($43.8 \pm 3.1$ mg vs. $42.0 \pm 3.9$ mg in normal and tumor tissues, respectively, Fig. 2a), the number of identified immunopeptides was greater in the tumor samples ($4378.5 \pm 335.8$ immunopeptides vs. $5463.2 \pm 367.2$ immunopeptides in normal and tumor tissues, respectively, Fig. 2b). We noticed that the protein concentration of the lysate was significantly higher in tumor samples ($1.1 \pm 0.1$ mg vs. $1.7 \pm 0.2$ mg in normal and tumor samples, respectively, Fig. 2c); therefore, the normalization of the identified number of immunopeptides by protein amount diminished the significance ($5365.1 \pm 901.2$ immunopeptides/protein vs. $4418.9 \pm 825.2$ immunopeptides/protein in normal and tumor samples, respectively, Fig. 2d). Similar to the protein amount, the relative quantity (RQ) of the α-chain obtained by Western blotting (Supplementary Fig. 7) was also higher in tumor samples (RQ of $0.50 \pm 0.05$ vs. $0.71 \pm 0.07$ in normal and tumor tissues, respectively, Fig. 2e); therefore, again, the normalization of the number of immunopeptides by RQ of the α-chain diminished the significance ($10,166.5 \pm 1268.2$ immunopeptide/RQ of the α-chain vs. $8502.2 \pm 783.3$ immunopeptide/RQ of the α-chain in normal and tumor tissues, respectively, Fig. 2f). Among these two factors (protein amount and the QR of the α-chain), the RQ of the α-chain exhibited a stronger correlation with the number of immunopeptides identified from tissue samples (Fig. 2g, h). There was no correlation between tissue weight and the number of immunopeptides identified (Supplementary Fig. 2c). Exact amounts and numerical scores can be found in Supplementary Data 2c. The ratio of nonbinders against deduplicated candidates in tissue samples was 8.4% on average (Supplementary Data 2e).

According to the analyses, a total of 44,815 unique immunopeptide were identified (6582 peptides per patient on average) (Fig. 2i). Among these, 5623 immunopeptides (12.5%) were normal-exclusively identified, and 14,049 immunopeptides (31.3%) were tumor-exclusive, while 25,143 immunopeptides (56.1%) were shared both in normal and tumor tissues (Fig. 2j, Supplementary Data 2d). The comparison identifying the number of immunopeptides and the overlap between normal and tumor tissues in identical patients can be found in Supplementary Fig. 8.

From the results, compared to previous reports on the analysis of solid tumor tissues such as colon[9], liver[15], and ovarian[14] cancer, the number of immunopeptides has been significantly improved when considering that the required amount of sample here is smaller. From this CRC immunopeptidome, among 11,117 source proteins (average 4149 proteins per patient, Fig. 2k), 817 (9.7%) or 1869 proteins (16.5%) were identified as normal or tumor-specific (Fig. 2l, Supplementary Data 2d). The identified immunopeptides exhibited the reasonable characteristics of HLA allotype-matched distribution and restraint by unsupervised clustering and binding prediction (Supplementary Fig. 9). The length of immunopeptide was variable according to the HLA allotype in each individual; nevertheless, the typical dominance of Class I immunopeptide, 9-mer in length, was clearly shown, and more than 99.7% of immunopeptides were between 8 and 12 amino acids in length (Supplementary Fig. 10). Notably, two neoantigens were identified from the tumor tissues of two distinct patients (Fig. 2m, n on the top spectrum; the lower spectrum obtained from the corresponding synthetic peptide for validation of identification exhibited the similarity of sample origin.) The first neoantigen identified from the primary tumor tissue of ID 172 carried a well-known CRC driver mutation, KRAS-G12V, at the position of the 7th to 16th amino acids [p. 7–16]. The other serine/threonine-protein phosphatase CPPED1 contains the R228Q mutation (CPPED-R228Q [226–234]) and was identified from ID 261's hepatic metastasized tumor tissue. KRAS-G12V is a representative cancer driver mutation that can induce cancer in various tissues. CPPED1 is also known as CSTP1, a negative regulator of Akt signaling, which directly dephosphorylates Akt at Ser473 and plays a role in carcinogenesis in cancers[20–22]. According to the motif information at IEDB and SYFPEITYHI (http://www.syfpeithi.de/0-Home.htm)[23], the neoantigen carrying KRAS-G12V was predicted to have a higher affinity against A*11:01 than the wild-type sequence due to the preferred amino acid substitution of valine at position 6 in the peptide (Fig. 2o, depicted as "Preferred: V" by an arrow). On the other hand, the CPPED1-R228Q mutation lost its deleterious amino acid arginine at position 3 by substitution with glutamine (Fig. 2p, depicted as "Deleterious: P/R" by an arrow). Therefore, compared with the corresponding wild-type (WT) sequence, the identified neoantigens showed two opposite patterns, that is, the acquisition of preferred residue or the loss of deleterious residue to increase the affinity against A*11:01. The same region of the peptide from wild-type origin was not identified from our sample. The background cause of how the mutation affected the affinity against the assigned HLA was seemingly different between the neoantigens derived from KRAS-G12V and CPPED1-R228Q, while both immunopeptides were predicted to have increased affinity compared to their wild-type counterparts by NetMHC prediction (Table 2).

**Comparison of immunopeptides from tissues revealed the cancer-specific profiles of immunopeptide trimming at pΩ.** For more insights from tissue-based immunopeptidomics, we next compared the trimming of the immunopeptide at the very last position of the C-terminus (pΩ) of immunopeptides. Among the overall immunopeptides identified, those peptides 8–12 amino acids in length (44,648 immunopeptides) and amino acids with a usage ratio of more than 0.5% are depicted in the pie chart (Fig. 3a, Supplementary Data 3a). Among these amino acids, cysteine had the lowest usage (0.5%). In contrast, tryptic (R, K) and chymotryptic (L, I, V, F, Y, W, A, M) amino acids were more common in the trimming of pΩ (Fig. 3a). Next, we compared the ratio (%) of each amino acid in trimming of immunopeptides at pΩ between normal and tumor immunopeptidomes within the

**Table 1 Binding predictions of identified neoantigens from HCT116 by NetMHCpan4.1 and NetMHC4.0.**

HCT116: HLA-A01:01,HLA-A02:01,HLA-B18:01,HLA-B45:01,HLA-C05:01,HLA-C07:01

| Gene | Genotype | Sequence | Assigned HLA by NetMHCpan4.1 | Bind Level by NetMHCpan4.1 | Assigned HLA by NetMHC4.0 | Bind Level by NetMHC4.0 | Affinity [nM] by NetMHC4.0 | Detection in HCT116 Immunopeptidome (10209) | Times Detected from 3 independent global analysis | In IEDB Epitope Catalog |
|---|---|---|---|---|---|---|---|---|---|---|
| AGO2 | WT | QEQKHTYLP | HLA-B45:01 | SB | HLA-B45:01 | WB | 1900.2 | + | 1 | + |
|  | H336Y | QEQKYTYLP | HLA-B45:01 | SB | HLA-B45:01 | WB | 980.7 | + | 1 | − |
| CHMP7 | WT | QTDQMVFNAY | HLA-A01:01 | SB | HLA-A01:01 | SB | 16.3 | + | 3 | + |
|  | A324T | QTDQMVFNTY | HLA-A01:01 | SB | HLA-A01:01 | SB | 33.5 | + | 3 | + |
| FNBP4 | WT | EEEKKGVAA | HLA-B18:01 | WB | n.p. | n.p. |  | + | 3 | + |
|  |  |  | HLA-B45:01 | SB | HLA-B45:01 | SB | 60.6 |  |  |  |
|  | K318E | EEEEKGVAA | HLA-B18:01 | WB | n.p. | n.p. |  | + | 3 | − |
|  |  |  | HLA-B45:01 | SB | HLA-B45:01 | SB | 83.2 |  |  |  |
| NAPA | WT | KAIDIYEQV | HLA-A02:01 | SB | n.p. | n.p. |  | − | 0 | + |
|  |  |  | HLA-C05:01 | WB | n.p. | n.p. |  |  |  |  |
|  | A181V | KVIDIYEQV | HLA-A02:01 | SB | HLA-A02:01 | WB | 50.1 | + | 3 | − |
|  |  |  | HLA-C05:01 | SB | HLA-C05:01 | WB | 6269.5 |  |  |  |
|  |  |  | HLA-C07:01 | WB | n.p. | n.p. |  |  |  |  |
| NR1D1 | WT | YSDNSNGSF | HLA-A01:01 | SB | HLA-A01:01 | SB | 33.0 | − | 0 | + |
|  |  |  | HLA-C05:01 | SB | HLA-C05:01 | SB | 5.0 |  |  |  |
|  |  |  | HLA-C07:01 | WB | n.p. | n.p. |  |  |  |  |
|  | G39D | YSDNSNDSF | HLA-A01:01 | SB | HLA-A01:01 | SB | 28.3 | + | 3 | + |
|  |  |  | HLA-C05:01 | SB | HLA-C05:01 | SB | 5.9 |  |  |  |
|  |  |  | HLA-C07:01 | WB | n.p. | n.p. |  |  |  |  |
| GAPDH | WT | AENGKLVIN | HLA-B45:01 | SB | HLA-B45:01 | WB | 892.6 | − | 0 | − |
|  | I69T | AENGKLVTN | HLA-B45:01 | SB | HLA-B45:01 | WB | 963.6 | + | 3 | + |
| IQGAP | WT | VLEDKVLSV | HLA-A02:01 | WB | HLA-A02:01 | SB | 35.1 | + | 2 | + |
|  |  |  | HLA-C05:01 | WB | HLA-C05:01 | WB | 5328.4 |  |  |  |
|  |  |  | HLA-C07:01 | SB | HLA-C07:01 | WB | 5137.6 |  |  |  |
|  | S1070T | VLEDKVLTV | HLA-A01:01 | WB | n.p. | n.p. |  | + | 1 | − |
| RBBP78 | WT | EERVINEEY | HLA-B18:01 | SB | HLA-B18:01 | SB | 103.2 | + | 3 | + |
|  | N17D | EERVIDEEY | HLA-B18:01 | WB | HLA-B18:01 | SB | 194.5 | + | 1 | + |
|  |  |  | HLA-B45:01 | WB | n.p. | n.p. |  |  |  |  |
| NOTCH2 | WT | NEGMCVTY | HLA-B18:01 | SB | HLA-B18:01 | SB | 6.4 | − | 0 | − |
|  | C41S | NEGMSVTY | HLA-B18:01 | SB | HLA-B18:01 | SB | 6.2 | + | 2 | + |
| PDP1 | WT | NEYTKFIPP | HLA-B18:01 | WB | HLA-B18:01 | WB | 626.2 | + | 3 | + |
|  |  |  | HLA-B45:01 | SB | HLA-B45:01 | WB | 2202.9 |  |  |  |
|  | N379D | DEYTKFIPP | HLA-B18:01 | SB | HLA-B18:01 | SB | 158.5 | + | 1 | − |
| UQCRB | WT | EEENFYLEP | HLA-B18:01 | WB | n.p. | n.p. |  | − | 0 | + |
|  |  |  | HLA-B45:01 | SB | HLA-B45:01 | WB | 863.2 |  |  |  |
|  | N88K | EEEKFYLEP | HLA-B18:01 | WB | n.p. | n.p. |  | + | 1 | + |
|  |  |  | HLA-B45:01 | SB | HLA-B45:01 | WB | 728.7 |  |  |  |

n.p.: not predicted as a binder.

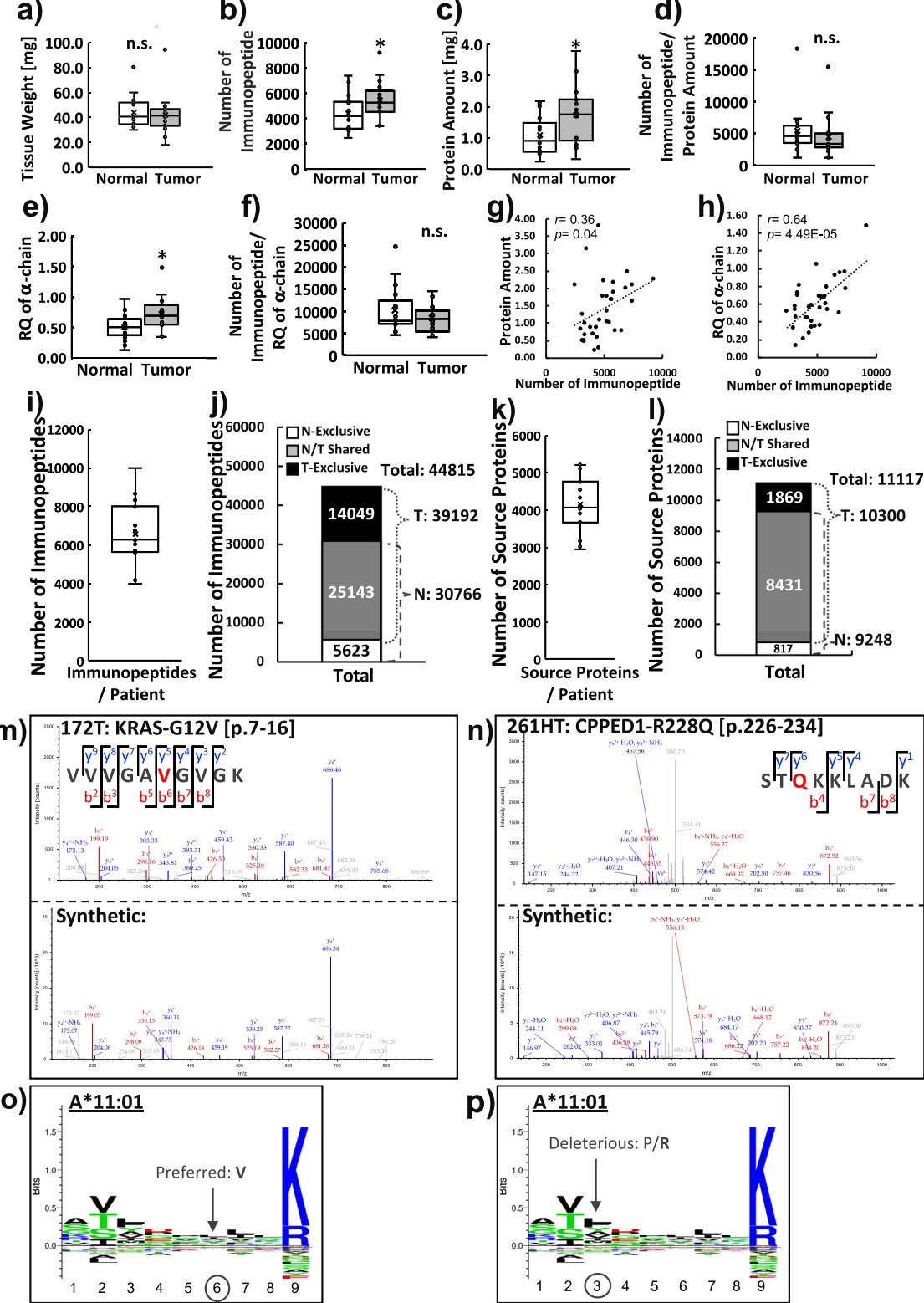

same individual. A comparison of samples that contained at least one or more amino acids of interest was included in the analyses. As a result, only trimming cysteine (pΩ-Cys) exhibited a statistically significant difference between the normal and tumor immunopeptidomes (13 out of 17 patients, $p = 0.0082$ by paired Student's $t$ test, Fig. 3b, Supplementary Data 3b). For more insights, the personal immunopeptidome was classified into three groups: immunopeptides only found in normal tissue (normal-exclusive), immunopeptides found in both normal and tumor tissues (shared), and immunopeptides only found in tumor tissue (tumor-exclusive) per patient. Then, the usage of amino acid frequency at pΩ was calculated as a ratio (%) against a subtotal of each group. Further breakdown of the personal immunopeptidome by exclusivity revealed an increase in pΩ-Cys

**Fig. 2 Personalized immunopeptidomics analyses from 17 colorectal cancer patients.** A global-immunopeptidomics analysis was performed on 17 CRC sample sets of normal and tumor tissues from identical patients. **a** The tissue weight used, **b** Number of immunopeptides identified, **c** protein amount used, **d** number of immunopeptides normalized by protein amount, **e** relative quantity (RQ) of α-chain amount, **f** number of immunopeptides normalized by RQ of α-chain, **g** correlation between number of immunopeptides identified and protein amount, and **h** correlation between number of immunopeptides identified and RQ of α-chain are shown in box plots with inter-quartile range (IQR). The center line inside the box indicates the median and the X indicates the average. n.s. no statistical significance, *$p < 0.05$ by unpaired Student's $t$ test. **i** The average number of identified unique immunopeptides per patient and **j** the total number of unique immunopeptides, i.e., the immunopeptidome of CRC (total unique immunopeptides) obtained from normal and tumor tissue samples are depicted. **k** The box plot depicts the average of the total number of source proteins per patient and **l** the total number of source proteins from all patients, identified by the same sample sets shown in **i**. The actually identified MS/MS spectrum of **m** KRAS-G12V [p.7–16]; VVVGAVGVGK and **n** CPPED1-R228Q [p.226–234]; STQKKLADK are depicted on top, and the lower panel shows the spectra acquired from the corresponding synthetic peptides for validation. Gray peaks are the background noise, which is probably derived from irrelevant precursor ions for identification. The amino acid substituted by somatic mutation within the neoantigen sequence is shown in red. NetMHCpan4.1 predicted that these two neoantigens are the possible binders of A*11:01. Backgrounds that are predicted to affect the increased affinity for A*11:01 by **o** gain of the preferred residue in KRAS-G12V and **p** the loss of the deleterious residue in CPPED1-R228Q are depicted.

immunopeptides in the tumor-exclusive population (Fig. 3d, Supplementary Data 3b) against the number of immunopeptides in each population (Fig. 3c, Supplementary Data 3b). The ratio of pΩ-Cys in the tumor-exclusive population (0.69 ± 0.14%) was statistically significantly higher than that of shared (0.36 ± 0.11%, $p < 0.001$) or normal-exclusive populations (0.33 ± 0.13%, $p = 0.019$, Fig. 3d). For other amino acids, the proportion of arginine-trimmed immunopeptides at pΩ (pΩ-Arg) exhibited a tendency of reduction in the tumor immunopeptidome (12 out of 17 patients, $p = 0.0576$ by paired Student's $t$ test, Fig. 3e, Supplementary Data 3c). Further breakdown of the personal immunopeptidome by exclusivity revealed a significantly lower ratio of pΩ-Arg in the tumor-exclusive population (11.29 ± 3.81%) than in the shared population (15.11 ± 5.17%, $p = 0.0315$, Fig. 3g, Supplementary Data 3c) against the number of immunopeptides in each population (Fig. 3f, Supplementary Data 3c). Although arginine and lysine are in the same tryptic group, these two amino acids exhibit different dynamics in immunopeptide presentation. For chymotryptic peptides, unexpectedly, the proportion of tryptophan-trimmed immunopeptides at pΩ (pΩ-Trp) exhibited a tendency to increase in the tumor immunopeptidome (13 out of 17 patients, $p = 0.0737$ by paired Student's $t$ test, Fig. 3h, Supplementary Data 3d). Further breakdown of the personal immunopeptidome by exclusivity revealed a significant increase in pΩ-Trp immunopeptides in the tumor-exclusive population (2.34 ± 0.76%) compared to the shared population (1.69 ± 0.54%, $p = 0.0336$, Fig. 3j, Supplementary Data 3d) against the number of immunopeptides in each population (Fig. 3i, Supplementary Data 3d). In the shift of pΩ trimming, there was a negative correlation between the ratio of pΩ-Trp in the normal- and tumor-exclusive immunopeptidomes ($r = -0.7846$, $p = 0.0003$, Fig. 3k, Supplementary Data 3d). The lower ratios of pΩ-Trp in the normal population correlated with higher ratios of pΩ-Trp in tumor tissue implied a suppressed processing of pΩ-Trp at normal condition and this can be released by cancer environment. These results indicated that DIM-MS immunopeptidomics analysis of clinical tissue samples is of use to reveal the unknown processing mechanisms of immunopeptides in the cancer microenvironment.

**Detection of oncogenic KRAS-carrying neoantigens by a global-immunopeptidomics approach.** The identification of neoantigens with KRAS-G12V mutation directly from clinical tissue (ID 172) implied that many neoantigens with oncogenic mutations have yet to be identified. To assess this hypothesis, we first analyzed the cell line Colo668, a small cell lung carcinoma cell line derived from a brain metastatic site, which has the same pair of KRAS-G12V with HLA-A*11:01 as ID 172, to determine whether the same neoantigen, KRAS-G12V [p. 7–16], would be

identified with reproducibility by a global-immunopeptidomics approach. By the same global-immunopeptidomics analyses from 1e8 Colo668 cells (a representative result of immunopeptidomics analysis from Colo668 for the CV distribution and the peptide overlap is shown in Supplementary Fig. 11), a total of 10,963 peptide groups were identified as candidates, and among these, 10,474 (95.5%) were predicted to be binders for the HLAs of Colo668 by NetMHCpan. Among these assigned immunopeptides, three neoantigens including KRAS-G12V [p. 7–16] were successfully identified (Table 3, Supplementary Fig. 12a). This KRAS-G12V [p. 7–16] identification was further verified by a targeted-MS approach, and a highly confident dopt score (>0.95) was obtained (Supplementary Fig. 12b). All three neoantigens were again predicted to have a stronger affinity than the corresponding wild-type sequences by NetMHC (Table 3).

Next, we analyzed the cell line RCM1, a rectal adenocarcinoma cell line with KRAS-G12V mutation without HLA-A*11:01, to determine whether any oncogenic KRAS-carrying neoantigens would be identified from different HLA allotypes. By the same global-immunopeptidomics analyses from 1e8 cells of RCM1 (a representative result of immunopeptidomics analysis from RCM for the CV distribution and the peptide overlap is shown in Supplementary Fig. 13), a total of 7534 peptide groups were identified as candidates, and among these, 7147 (94.9%) were predicted as binders for the HLAs of RCM1 by NetMHCpan. Among these assigned immunopeptides, a sequence of neoantigen, KRAS-G12V [p.11–19] (AVGVGKSAL), was successfully identified by global-immunopeptidomics analysis (Table 3, Supplementary Fig. 14a). This KRAS-G12V [p. 11–19] identification was further verified by a targeted-MS approach, and a highly confident dopt score (>0.95) was obtained (Supplementary Fig. 14b). Another independent global-immunopeptidomics analysis of the RCM1 cell line again identified KRAS G12V [p. 11–19] with reproducibility from a total of 7662 peptide groups identified. As a result, a total of 9448 peptide groups were identified from the RCM1 cell line, and among these, 8914 (94.3%) were predicted as binders for the HLAs of RCM1 by NetMHCpan. The affinity prediction for KRAS-G12V [p. 11–19] for HLA-C*01:02 by NetMHC was not available because of the limited coverage of HLA molecules with respect to NetMHC. These results indicated that DIM-MS is of use in global immunopeptidomics to identify potential shared neoantigens from various scarce materials.

## Discussion
In this study, we established a highly efficient immunopeptidomics method allowing for the comparative analysis of the normal- or cancer-exclusive immunopeptidomes from the same individual, which revealed cancer-specific signatures of immunopeptides at

**Table 2 Binding predictions of neoantigens from colorectal tissues by NetMHCpan4.1 and NetMHC4.0.**

ID172T: HLA-A11:01,HLA-A24:02,HLA-B35:01,HLA-B40:02,HLA-C01:02,HLA-C03:03

| Gene | Mutation | Sequence | Assigned HLA by NetMHCpan4.1 | Bind Level by NetMHCpan4.1 | Assigned HLA by NetMHC4.0 | Bind Level by NetMHC4.0 | Affinity [nM] by NetMHC4.0 | Detection in ID172T Immunopeptidome (7067) | In IEDB Epitope Catalog |
|---|---|---|---|---|---|---|---|---|---|
| KRAS | WT | VVVGAGGVGK | HLA-A11:01 | WB | HLA-A11:01 | WB | 299.7 | - | + |
|  | G12V | VVVGAVGVGK | HLA-A11:01 | SB | HLA-A11:01 | WB | 137.3 | + | + |

*Prediction against C*01:02 is not available by NetMHC4.0.

ID261HT: HLA-A11:01,HLA-A24:02,HLA-B13:01,HLA-B52:01,HLA-C07:02,HLA-C12:02

| Gene | Mutation | Sequence | Assigned HLA by NetMHCpan4.1 | Bind Level by NetMHCpan4.1 | Assigned HLA by NetMHC4.0 | Bind Level by NetMHC4.0 | Affinity [nM] by NetMHC4.0 | Detection in ID261HT Immunopeptidome (8652) | In IEDB Epitope Catalog |
|---|---|---|---|---|---|---|---|---|---|
| CPPED1 | WT | STRKKLADK | HLA-A11:01 | WB |  | n.p. |  | - | - |
|  | R228Q | STQKKLADK | HLA-A11:01 | SB | HLA-A11:01 | SB | 52.2 | + | - |

n.p.: not predicted as a binder by NetMHC4.0.
*Prediction against B*13:01, B*52:01 and C*12:02 are not available by NetMHC4.0.

the tissue level. Next, we further demonstrated the presentation of more variation of potential shared neoantigens carrying oncogenic KRAS.

It has been a long-standing challenge to identify neoantigens directly from small clinical tissues: our personalized immuno-peptidomics analysis achieved that task and exhibited a potential to reveal the distinct populations of immunopeptides in cancer tissue across patients. Intriguingly, unlike previous reports, the direct identification of neoantigens with a representative cancer driver mutation, an oncogenic KRAS, from the terminal stage of colon cancer tissue was achieved by an established DIM-MS-based global-immunopeptidomics approach.

In the past, immunopeptidomics analysis required a large number of samples, especially from clinical tissues[12,14,15]. Alternatively, patient-derived cancer cells must be prepared as primary cultures or culture organoids to ensure a sufficient amount of sample[17,24]. The purified cultures of patient-derived cancer cells are convenient for various analyses. However, there is a possibility that the time-consuming processes of cell culture without a cancer microenvironment may affect the presentation of immuno-peptides and cause differences versus the original state. Although the mouse xenograft model can also secure patient-derived cancer cells, the response to ICIs may vary depending on the host organism, as shown through the discrepancy in the results of clinical trials of MEK inhibitors[25] with PD-L1 treatment[26]. Therefore, it is very complicated to evaluate the efficacy of cancer immunotherapy through model materials. FAIMS-assisted DIM-MS paved the way for the identification of ~5000 immunopeptides from 40 mg of tissue (a cube 3–5 mm in size) without the need for bias-prone chemical prefractionation procedures[27]. Thus, our global-immunopeptidomics approach is an ideal method for direct and robust analysis from scarce materials.

In principle, a higher mutation load means more possible neoantigens and is considered to be associated with better outcomes of ICI treatment[28,29]. However, despite the highest mutation load in sample ID 119 (1752, Table 2b), no neoantigen was detected, even from a total of 6983 immunopeptides. According to previous in vitro studies, IFN-γ stimulation induces the immunoproteasome, which facilitates more preferred immunopeptide trimming for HLA class I immunopeptide presentation[30], resulting in an increased presentation of immunopeptides[9,31]. Therefore, the presentation of immuno-peptides is considered to be increased in the cancer micro-environment under IFN-γ stimulation. However, there have been few reports of direct comparisons of the immunopeptidomes between normal individuals and cancer patients from the same individual. In this study, as with mutation load, the number of immunopeptides identified in cancer tissue was not significantly different from the number identified in normal tissues when normalized by rate limiting factors (Fig. 2d, f). These results indicate that the previous studies in immunopeptidomics based on purified cell mass or patient-derived xenografts might have reflected only a part of the immunopeptidome due to the lack of a cancer microenvironment.

In previous immunopeptidome analyses using in vitro samples with or without IFN-γ stimulation, it has been reported that chymotryptic trimming (mainly by leucine, isoleucine, and valine) becomes dominant in response to the induction of the immunoproteasome[12,17]. However, of the amino acids targeted by chymotrypsin, only the immunopeptides trimmed by trypto-phan exhibited a tendency to increase in tumor tissue. Further breakdown revealed that this was attributed to the increase in trimming by tryptophan at pΩ, especially in tumor-exclusive immunopeptides. It has been reported that chronic exposure to IFN-γ is known to induce skipping/frameshifting translation at

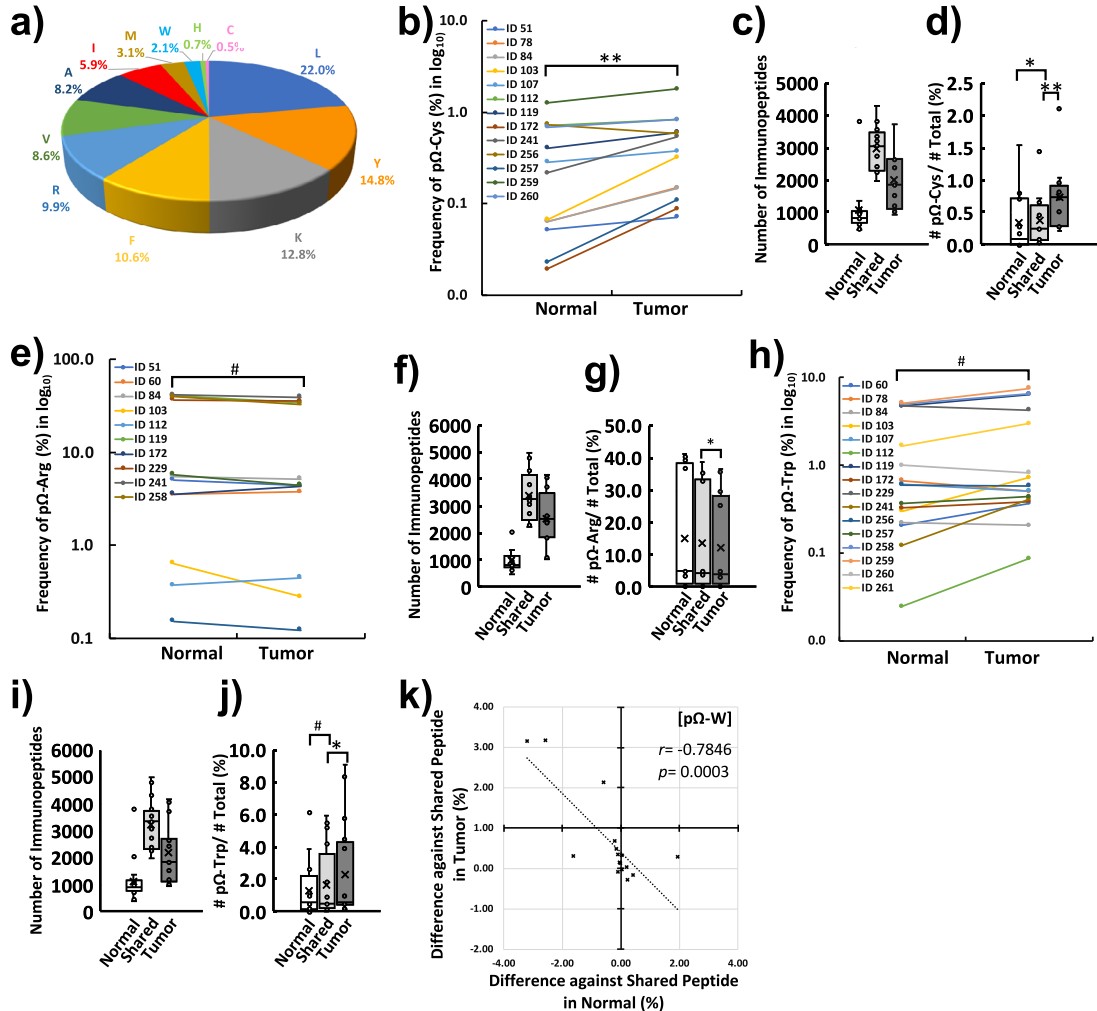

**Fig. 3 The CRC immunopeptidome, neoantigens directly identified from tissues and the cancer-specific profiling of HLAp by global immunopeptidomics analysis. a** Frequency of amino acid usage at the C-terminus (pΩ) of immunopeptides identified from CRC samples. Immunopeptides eight to 12 amino acids in length (total 44,648) and exceeding 0.5% frequency are depicted in the pie chart. **b** A trend-shift of cysteine at pΩ (pΩ-Cys) in the tumor immunopeptidome within CRC patients. Only pΩ-Cys immunopeptide-identified samples ($n = 13$) are depicted. The percentile frequency is depicted with a logarithmic (log10) vertical axis. **$p < 0.01$ by paired Student's $t$ test. **c** The box plot indicates the number of immunopeptides in the group of normal-exclusive, shared and tumor-exclusive labeled with Normal, Shared and Tumor, respectively. **d** The box plot indicates the frequency of pΩ-Cys immunopeptides in each groups. Significance of frequency is depicted as *$p < 0.05$, **$p < 0.01$ by Student's paired $t$ test. **e** A trend-shift of arginine at pΩ (pΩ-Arg) in the tumor immunopeptidome within CRC patients. A result from pΩ-Arg immunopeptide-identified samples ($n = 12$) is depicted. The percentile frequency is depicted with a logarithmic (log10) vertical axis. The normal-exclusive population includes more pΩ-Arg immunopeptides. #$p = 0.058$ by paired Student's $t$ test. Further breakdown of immunopeptides by exclusivity revealed significantly lower pΩ-Arg immunopeptides in the tumor-exclusive population. **f** The box plot indicates the number of immunopeptides in the group of normal-exclusive, shared and tumor-exclusive labeled with Normal, Shared and Tumor, respectively. **g** The box plot on the right indicates the frequency of pΩ-Arg immunopeptides in each groups. *$p < 0.05$ by paired Student's $t$ test. **h** A trend shift of tryptophan at pΩ (pΩ-Trp) in the tumor immunopeptidome within CRC patients. Only pΩ-Tryp immunopeptide-identified samples ($n = 16$) are depicted. The percentile frequency is depicted with a logarithmic (log10) vertical axis. The tendency of frequency is depicted as #$p = 0.074$ by paired Student's $t$ test. Further breakdown of immunopeptides by exclusivity revealed the tendency of increased frequency of pΩ-Trp immunopeptides in the tumor-exclusive population. **i** The box plot indicates the number of immunopeptides in the group of normal-exclusive, shared and tumor-exclusive labeled with Normal, Shared and Tumor, respectively. **j** The box plot on the right indicates the frequency of pΩ-Trp immunopeptides in each groups. *$p < 0.05$, #$p = 0.080$ by Student's paired $t$ test. **k** Pearson correlation coefficient ($r$) revealed the negative correlation of the frequency of pΩ-Trp immunopeptides between normal-exclusive and tumor-exclusive samples from identical patients. The lower the frequency of pΩ-W was in the normal-exclusive population, the higher the frequency of pΩ-W was in the tumor-exclusive population.

certain tryptophan codons that leads to aberrant peptide translation. In addition, these aberrant peptides have been reported to become a source for neoantigens in melanoma cells[32]. A significant increase in immunopeptides trimmed by cysteine in tumor samples compared to normal samples is also of note; although the trend is slight, the cysteine association in immunopeptide processing has been underrepresented in the past[33]. The tumor-unique profiling of immunopeptides, such as

increased trimming by cysteine and tryptophan and the underlying physiological mechanisms, as well as the associations with clinical outcomes, will be clarified in the future. Since the allele frequency of HLA is known to differ according to race, the combination of HLA alleles in individuals is further complicated. Under such complexity, to delineate a clinically relevant signature with more contrast, the immunopeptidome from the HLA allotype-matched normal/tumor immunopeptidome from the

**Table 3 Binding predictions of identified neoantigens from Colo668 and RCM1 by NetMHCpan4.1 and NetMHC4.0.**

Colo668: HLA-A01:01,HLA-A11:01,HLA-B15:17,HLA-B44:02,HLA-C07:01,HLA-C07:04

| Gene | Mutation | Sequence | Assigned HLA by NetMHCpan4.1 | Bind Level by NetMHCpan4.1 | Assigned HLA by NetMHC4.0 | Bind Level by NetMHC4.0 | Affinity [nM] by NetMHC4.0 | Detection in Colo668 Immunopeptidome (10474) | In IEDB Epitope Catalog |
|---|---|---|---|---|---|---|---|---|---|
| KRAS | WT | VVVGAGGVGK | HLA-A*11:01 | WB | HLA-A11:01 | WB | 299.7 | − | + + |
|  | G12V | VVVGAVGVGK | HLA-A*11:01 | SB | HLA-A11:01 | WB | 137.3 | + | + + |
| CERT | WT | AIIIYQTHK | HLA-A*11:01 | SB | HLA-A11:01 | SB | 40.7 | − | − |
|  | I489V | AVIIYQTHK | HLA-A*11:01 | SB | HLA-A11:01 | SB | 32.0 | + | − |
| MDPI | WT | RYFVHREIY | HLA-C*07:01 | SB | HLA-C*07:01 | SB | 469.5 | − | − |
|  | I95T | RYFVHRETY | HLA-C*07:01 | SB | HLA-C*07:01 | SB | 282.0 | + | − |

*Prediction against C*07:04 is not included because of prediction unavailability by NetMHC4.0.

RCM1: HLA-A24:02,HLA-B51:01,HLA-B54:01,HLA-C01:02,HLA-C14:02

| Gene | Mutation | Sequence | Assigned HLA by NetMHCpan4.1 | Bind Level by NetMHCpan4.1 | Assigned HLA by NetMHC4.0 | Bind Level by NetMHC4.0 | Affinity [nM] by NetMHC4.0 | Detection in RCM1 Immunopeptidome (8914) | In IEDB Epitope Catalog |
|---|---|---|---|---|---|---|---|---|---|
| KRAS | WT | AGGVGKSAL | n.p. |  |  | n.a.* |  | − | − |
|  | G12V | AVGVGKSAL | HLA-C*01:02 | WB |  | n.a.* |  | + | − |

n.p.: not predicted as a binder.
*Prediction against C*01:02 is not included because of prediction unavailability by NetMHC4.0.

same individual is thought to be indispensable. From the perspective of establishing signature profiling analyses from in-depth individual immunopeptidomes, DIM-MS with different search methods, such as customized aberrant peptide database searches[32] or de novo MS analysis, should be beneficial[13,34]. Further analyses are required to clearly delineate the new aspects of processing/presentation of immunopeptides. The dataset of individual immunopeptidomes from both normal and cancer obtained in this study will be of use for those who seek to explore the new landscapes in immunopeptidomics.

Due to the eradication of neoantigen-presenting cancer cells by antitumor immunity[31,35], it has been suggested that the frequency of neoantigen presentation on solid tumors might be low[9,15,17], especially in CRC[5,6]. However, in this study, 2 neoantigens, one of which carried KRAS-G12V, were directly identified from stage IV CRC tissue. Recently, the affinity of two KRAS-G12V-carrying neoantigens for HLA-A*03:01 or A*11:01 has been quantitatively characterized by PRM using KRAS-G12V-transduced monoallelic cell samples[19]. Although the monoallelic expression of HLA is a useful experimental model to characterize the affinity and the restraint of immunopeptides, the actual presentation of immunopeptides may vary due to the similarity of binding motifs between intrinsic HLAs. In this study, neoantigens were identified under fully allelic conditions. This implies that it is very likely that most of the potential neoantigens carrying cancer driver mutations have yet to be identified. Indeed, another variation of neoantigen with oncogenic KRAS was identified from RCM1, a commercially available colon cancer cell line. The identified neoantigen of oncogenic KRAS-G12V [p.11–19] was predicted to be a weak binder for HLA-C*01:02 by NetMHCpan, while affinity prediction by NetMHC was not available due to the limited HLA coverage in the algorithm. In some neoantigens from HCT116, e.g., FNBP4-K318E for HLA-B*45:01 and NAPA-A181V for HLA-C*05:01, C*07:01, prediction gaps between NetMHCpan and NetMHC were observed. The actual experimental affinity measurements for these inconsistent immunopeptides will also be critical to accomplish better prediction. These results indicate that the incomplete prediction algorithm will eventually erroneously predict some possible binders as nonbinders. Owing to the wide variety of binding motifs of a vast number of HLA types, more time will inevitably be required to build a complete prediction algorithm. The previous preprint version of this manuscript included the KRAS-G13D carrying neoantigen from HCT116 by the targeted-MS screening approach, while the dotp score of this neoantigen (dotp = 0.90 by targeted-MS) did not achieve the recommended threshold (dotp = 0.91 or more) and thus excluded the data from this article. The identification of immunopeptides by targeted-MS without fractionation makes identification more complex, and a more refined methodology will be needed. Targeted-MS with FAIMS is also an attractive option, considering the merit of fractionation from scarce sample sources. However, the optimal CV for an immunopeptide of interest varies according to peptides. For example, in this study, the corresponding synthetic peptide of KRAS-G12V [p. 7–16] was shown to have a relatively broad range of CV for neoantigen-identifying PSMs from CV −70 to −35 V (Supplementary Fig. 15a–f), while the actual neoantigen in tissue samples (ID172T and Colo668) was identified from the more restricted CV range from −60 to −65 V (Supplementary Fig. 15g). As such, most neoantigens in the actual sample exhibited a more restricted range of CV. Since the actual sample contains various competing peptides, it is thus too difficult to establish the FAIMS-assisted targeted-MS method without the knowledge of optimal CV(s), at least at present, or which CV to choose for the neoantigen of interest. Given this situation, listing the shared neoantigens from various samples first and

foremost by global-immunopeptidomics is more of use to establish the panel screening of shared neoantigens as a companion diagnostic, possibly by future FAIMS-assisted targeted-MS.

Even without information on actually presented neoantigens, immunotherapy with autologous T cells that recognize KRAS-G12D-carrying neoantigens in lung-metastasized CRC has been reported to be effective and successful in a previous clinical trial[36]. However, considerable time and cost are required to screen useful neoantigens from predicted candidates. Identifying neoantigens by mass spectrometry may mitigate this health economics issue. It is expected that established global-immunopeptidomics analysis applicable for scarce clinical materials by FAIMS-assisted DIM-MS will grant better knowledge both in the basic and clinical aspects of immunopeptidomics, further enlightening us with a new outlook for precision cancer medicine.

## Methods

**Cell lines and clinical tissue samples**. The cell lines used in this study are described in the Supplementary Methods. All surgically removed clinical tissues were dissected into the desired size and immediately cryopreserved until use. Approximately 40 mg of net weight tissue was subjected to immunoaffinity purification. Among the obtained tissue samples, 15 out of 17 were the set of primary colon tumor tissue with normal colon tissue. The other 2 of 17 were the set of liver-metastasized colon tumor tissue with normal colon tissue.

**Ethics approval**. Metastatic colon cancer tissues at pathological stage IV and adjacent normal tissues were obtained from patients who provided written informed consent for this study. This study was first approved by the ethical committee of the Japanese Foundation for Cancer Research (JFCR) (Ethical committee number 2010-1058) and has been periodically renewed.

**Immunoprecipitation of HLA-Class I complex and immunopeptide purification**. The in-house purified anti-panHLA antibody (W6/32) was used for immunoprecipitation of the HLA complex. Detailed information on sample preparation for immunopeptidomics can be found in the Supplementary Methods. Briefly, the desired amount of cell pellet or tissue was lysed by 1 mL of lysis buffer containing [20 mM] HEPES, [150 mM] NaCl, 1% NP-40, 0.1% SDS, and 10% glycerol on ice. Iodoacetamide at a final concentration of 0.2 mM and protease inhibitor cocktail (Halt™) were added to lysis buffer just before lysate preparation. After centrifugation for clarification, the supernatant was subjected to immunoprecipitation with 800 μg W6/32 antibody cross-linked to 200 μl of Protein G Sepharose beads overnight at 4 °C on a rotating rotor. After the washing steps, the trimer of the HLA complex was dissociated and eluted with 500 μl of 1% TFA. The obtained eluate was further processed by tC18 SepPak (Waters), and the immunopeptide was enriched by 500 μl of 20% acetonitrile. The prepared immunopeptide samples were then dried by an evaporator and stored at −30 °C until MS analysis.

**Construction of a customized database for cell lines**. The customized databases for cell lines were established by whole-exome sequencing (WES) analysis from the COSMIC Cancer Cell Line Project. All listed somatic mutations were processed by an in-house built pipeline as previously reported[37,38]. Then, the obtained cell line-specific mutations were combined with the Swiss-Prot human proteome database (20,181 entries) from the UniProt website to perform the customized database search.

**HLA typing, WES, and in-house pipeline for personalized databases of clinical tissues**. For clinical tissue samples, genomic DNA was extracted from normal and tumor tissues by a QIAmp DNA mini Kit (QIAGEN). To establish the personalized database for global-immunopeptidomics, WES analysis was performed in all samples as previously described[38,39]. Briefly, the genome libraries for WES were established from the DNA samples using the xGen Exome Research Panel (IDT, Coralville, IA) according to the manufacturer's instructions. The exons were sequenced as 150 base pair paired-end reads by a NovaSeq6000 system (Illumina, San Diego, CA). The sequence data obtained were then analyzed to select possible germline variations and somatic mutations. In short, sequences of the whole exome were compared with a reference human genome (hs37d5), and the possible germline variants and somatic mutations were identified. The variants in public databases (>1%) were also excluded. Then, cancer somatic mutations were extracted by subtraction of germline variation identified in normal tissue from tumor tissue. Among those mutations, only the missense mutations with amino acid substitution were selected and translated into full-length protein and added to the same SwissProt/UniProt human database described above as a personalized database. The cancer-specific mutations in individual patients were extracted by subtraction of nonrelevant (germ line) mutations identified in normal tissue. The

established personalized database was used for both normal and tumor immunopeptidomics corresponding to patient samples. OptiType was employed for 4-digit HLA typing from deep sequencing data. The genetic details, including mutation burden, oncogenic KRAS mutations, and the HLA genotypes of clinical samples, can be found in Supplementary Data 1b.

**Global-immunopeptidomics by high-field asymmetric-waveform ion mobility spectrometry (FAIMS)-assisted differential ion mobility (DIM) mass spectrometry (MS)**. The full methodology for global-immunopeptidomics analyses can be found in the Supplementary Methods. Briefly, by using the Ultimate 3000 RSLC nano HPLC system, the immunopeptide samples were first trapped by a precolumn (PepMap, Thermo Fisher Scientific) and then separated by an analytical column (Aurora, IonOpticks). For seamless gas phase fractionation[40], the FAIMS-Pro interface (Thermo Fisher Scientific) was installed onto an Orbitrap Fusion Lumos Tribrid mass spectrometer (Thermo Fisher Scientific) to perform the DIM-MS. For global-immunopeptidomics, a total of three independently run CV sets were introduced for each sample. Each CV set included three different CVs, so a total of nine fractions from three analyses were obtained per sample. All acquired files were searched against the corresponding customized/personalized database.

**Database search for global-immunopeptidome**. The obtained raw data were processed by Proteome Discoverer version 2.4 using the Sequest HT engine against the corresponding database. For the enzymatic designation, the search was set to no-enzyme (unspecific). The precursor mass tolerance was set to 10 ppm, and the fragment mass tolerance was set to 0.6 Da. The maximum number of missed cleavages was set to zero. Methionine oxidation and cysteine carbamidomethylation were included as dynamic modifications. For percolator filtering, the false discovery rate (FDR) was set to 0.01 (Strict) to identify highly confident peptides. The output of peptide groups was exported as a list in Excel format. The peptide sequences in the list were first deduplicated only for unique sequences and then filtered by peptide length from 8- to 15-mer as candidate immunopeptides. Finally, the sequences predicted as no-binder with default settings (% rank threshold for strong binders as 0.5% and weak binders as 2%) by NetMHCpan 4.1[41] were further excluded from the list to determine the assigned HLA. The affinity between immunopeptides and the corresponding HLA allotype was obtained by NetMHC4.0 with the default settings (% rank threshold for strong binders as 0.5% and weak binders as 2%). Simultaneously, for unsupervised clustering of immunopeptide and motif sequences, Gibbs Cluster 2.0[42] and Seq2 logo[43] were used to confirm whether the identified immunopeptide/motifs were consistent with the HLA allotypes according to each sample. The preferred and deleterious amino acids within HLA-A*11:01 can be found in SYFPEITHI[23] and the immune epitope database (IEDB) and analysis resource website (https://www.iedb.org/mhc/213).

**Profiling of immunopeptide in normal and tumor tissues**. To investigate the possible distinctions between normal and tumor-derived immunopeptides, the intersection of immunopeptides between normal and tumor tissue was calculated, and Venn diagrams were generated by a publicly available website (http://bioinformatics.psb.ugent.be/webtools/Venn/). Based on the Venn diagrams, the personal immunopeptidome was classified into the following 3 groups: (1) normal-exclusive, (2) shared in both normal and tumor tissues (here called as shared-peptide) and (3) tumor-exclusive. The frequency of the usage of each amino acid at the C-terminus of immunopeptide (pΩ) was calculated as a percentage (%) relative to the total number of immunopeptides in each group. The difference in the frequency of usage of the amino acid at the C-terminus between the normal-exclusive group and the tumor-exclusive group was calculated by comparison with the frequency of the shared-peptide group.

**Targeted immunopeptidomics by parallel reaction monitoring (PRM)**. The oncogenic KRAS status for cell lines was available from THE RAS INITIATIVE at the National Cancer Institute (NCI) of the National Institutes of Health (NIH) (https://www.cancer.gov/research/key-initiatives/ras/outreach/reference-reagents/cell-lines). Except for the uninstalled condition of the FAIMS-Pro interface, the preparation of samples and the conditions of MS instruments were in accordance with the same settings as the abovementioned global-immunopeptidomics. For the data acquisition of the targeted-immunopeptidomics, the PRM strategy was introduced with minor modifications[44]. Briefly, collision CID and the detector ion trap were used in rapid scan mode. The resolution was set to 15,000, and the maximum ion injection time was 300 ms. To prepare the inclusion list, the predicted m/z was calculated by ChemCalc (https://www.chemcalc.org/peptides)[45] for the immunopeptide of interest. The obtained raw data were searched against the database that includes the full-length amino acid sequences of corresponding oncogenic KRAS and the source protein of the positive controls by Proteome Discoverer 2.4. The filtering was performed by target-decoy search[46] with 1% FDR.

**Statistics and reproducibility**. All data are expressed as the mean with standard error (mean ± SE). Unpaired Student's t test was used for statistical significance analysis, except for the assessment of the shift in immunopeptide trimming at pΩ that was validated by paired Student's t test. The Pearson correlation coefficient was used to assess the correlation analysis. Statistical significance thresholds were

determined at \*$p < 0.05$, \*\*$p < 0.01$, and \*\*\*$p < 0.001$ values. For the validation of FAIMS-assisted global-immunopeptidomics, more than three independent IP samples were prepared from HCT116 cells and analyzed in injection triplicates. All experiments were replicated and are reproducible except for the clinical tissue samples. Due to sample scarcity, global immunopeptidomics with 3 CV sets was intensively performed for clinical tissue samples.

**Reporting summary**. Further information on research design is available in the Nature Research Reporting Summary linked to this article.

## Data availability

The LC/MS raw data, summarized result files and the list of identified immunopeptides from samples have been deposited into a public open access proteomic database, the Japan Proteome Standard Repository/Database (jPOST), as follows: HCT116 immunopeptide without FAIMS in JPST001072, HCT116 immunopeptide with FAIMS in JPST001066, global identification of HCT116 immunopeptide in JPST001068, immunopeptide from normal regions of CRC tissues in JPST001070, and immunopeptide from tumor regions of CRC tissues in JPST001069. For CRC tissue samples, the associating whole-exome sequencing data has been deposited into JPST001070 and JPST001069 respectively. The images of uncropped and unedited blots can be found in Supplementary Fig. 16. All source data underlying the graphs and charts presented in the main figures is available as Supplementary Data 1–3.

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

## Acknowledgements

This work was supported by the Development of Technology for Patient Stratification Biomarker Discovery of the Japan Agency for Medical Research and Development (20ae0101074s0302) and Grant-in-Aid for Scientific Research (C) of the Japan Society for the Promotion of Science (20K05759).

## Author contributions

Conceptualization: Y.M. and K.U. Methodology: Y.M., Y.H., R.F., N.S., and K.U. Genomic Analysis and Tailored Database: K.K., K.N., and N.S. Clinical Sample Collection: Y.I., S.N., and R.F. Experiments and Data Curation: Y.M. and K.U. Original Draft: Y.M. and K.U.

## Competing interests

The authors declare no competing interests.
