## [Peer Review File · Communications Biology]

Reviewers' comments:

Reviewer #1 (Remarks to the Author):

Review manuscript COMMSBIO-21-3009 T

Minegishi and colleagues describe a FAIMS assisted immunopeptidomics discovery workflow that yields increased numbers of peptides compared to a no FAIMS approach. After evaluating data acquisition strategies on an HCT-116 sample, the authors then employ FAIMS acquisition on a panel of 17 CRC tissues with matched normal. They identify over 40,000 peptides including 2 neoantigens and detect tumor specific antigen processing. They furthermore develop a targeted PRM assay for KRAS derived neoantigens and validate the feasibility in a set of colorectal cancer cell lines. This study employs an innovative data acquisition approach and is of general interest to the field of immunopeptidomics. However, the authors should address the following concerns in a revised version before I can recommend publication of the manuscript.

Major concerns:

- The authors describe a relatively high yield of >3,000 peptides from as little as 5e6 cells. For Fig 1b and following, can the authors clarify the input amount and combination of CVs used to achieve this number. It is not clear from the method section whether an equivalent of 5e6 cells was injected 3 times for 9 CVs or whether this number is derived by adding up 3 injections of 5e6 each to a total of 1.5 e7 cells. Is the no FAIMS comparison also performed using 3 injections?
- What is the overlap of peptides across 9 CVs and which CVs yield the most peptides? What was the cycle time or TopN method used per CV experiment during data acquisition? The vendor recommends 1 additional CV per additional hour of measurement time. Here, 3 CVs are used during a 1 hour gradient. Could one reduce analysis time and increase yield by reducing the CVs sets and the number of injections? Particularly for low input samples, could the yield be increased further by selecting a longer gradient/fewer CVs. Have the authors compared their FAIMS settings with previously published methods for immunopeptidome analysis using FAIMS.
- The authors have uploaded the data to the Japan Proteome Standard Repository/Database (jPOST), however no reviewer login details were provided. Therefore, the quality of the raw data and claims made in the publication could not be evaluated and confirmed. Please provide the raw data to reviewers before publication.
- Minegishi et al observe a cancer specific tryptophan trimming of peptides. The trend is already very small, could this not just be due to comparing 5,603 normal exclusive to 14,052 tumor specific peptides? Is this trend also observable when including the shared peptides between tumor and normal tissue?
- The authors also observed increased cysteine at the C-terminus of tumor exclusive peptides. As also mentioned in the text, cysteine has not been observed as anchor residue for HLA-I alleles and other studies analyzing HLA bound peptides by mass spectrometry reported poor recovery of cysteines overall. Taken together, this could also indicate that peptides with a C-terminal cysteine are more likely false positive identifications. Did the authors reduce and alkylate to increase yield of cysteine containing peptides?
- This study also describes the development of a KRAS neoantigen specific PRM assay. How many cells were used for the detection of neoantigens in the cell lines studied? It would be interesting to see the results of a PRM analysis using the CRC tissue samples. Could this increase the number of neoantigens discovered for this tumor?

Minor concerns:

- The authors comment that true binding peptides might be missed by current prediction algorithms, however, in the methods they state that their analysis is restricted to peptides that predict as binders. How many peptides would they detect if not filtering for predicted ones?
- From the method section, it is not apparent how much antibody or beads crosslinked with antibody were used per IP.
- The authors use a stepped elution approach with 20% and 80% acetonitrile. Are both fractions analyzed on the MS or only the 20% fraction? Do the authors think they would lose peptides by not eluting at higher concentrations such as 30-50% as published by others in the field?
- Could the use of FAIMS benefit the sensitivity of the PRM assay further? Are the same neoantigens detected within the same CV?
- Please copy edit the manuscript for readability to convey the correct meaning of each sentence. As example, line 70 should probably read indispensable instead of dispensable.

Reviewer #2 (Remarks to the Author):

I have had the pleasure to review the article entitled "Comprehensive-immunopeptidomics analysis reveals presentation of potential neoantigens carrying cancer driver mutations" by Koji Ueda and his group. This group is world-renowned for their work in this field and among the leaders in this field, which is again proven with this work.

I wish to congratulate the authors to their study and the intent to improve HLA-ligandomics with new approaches to yield more data and better insights through additional peptide identifications. This is important work and should be commended. The amount of >6,000 HLA ligands they are able to characterize in minute amounts of samples is truly impressive. Further the combination of different mass spectrometry approaches, e.g. comprehensive analyses and targeted mass spectrometry is a great feature of this presented work.

Obviously, the central aim of their endeavor is to characterize mutated HLA ligands in small tissue samples, which is potentially relevant for individualized therapy approaches. Further they identify mutations in frequent driver mutations (here KRAS in colorectal cancer) that are repeatedly mutated and shown in different patients.

Major issues

My one central and only major concern to this manuscript is the spectra of the mutated HLA-ligands as provided in Fig. 2 and the supplements.

I would like to see some way of validation the presented fragmentation spectra are actually true.

For some examples that may prevent confidence, in spite of the strict false discovery rate set to 0.01 to identify peptides:

In Fig. 2 C (left lower graph) I can hardly discern the largest fragment at 500.70 m/z (I think), which remains unassigned. Further there are a variety of other small peaks in mutCPPED1 that are unexplained.

The mutKRAS sequence in Fig. 2c is extremely repetitive, since it is constituted from 5 of 10 amino acids that are actually all valine in the peptide. This may be an issue since this low diversity may favor false discoveries.

In suppl. Fig. 7 the spectra are very busy as well and the largest peak of GAPDH 463.97 m/z remains unexplained, next to two other unassigned peaks. The same goes for CHMP7, where 615.17 m/z remains without explanation.

Furthermore, when comparing mutKRAS peptide with the G13D variant (suppl. Fig 7d & 7e), suggested to be repetitive spectra, we can discern a 357.32 m/z and 357.37 m/z peak respectively. The first mentioned peak being the peak with the highest intensity remains unexplained and although generally comparable, for instance regarding assigned peaks the γ 9 2+ peak 451.95 m/z is the largest peak in e), whereas we only find a small γ 9 2+ peak in d) with 451.80 m/z, seems relevantly different.

The spectra absolutely might be what the authors claim them to be.

Nevertheless, there needs to be some kind of validation for sufficient confidence.

E.g. one could think of a synthetic peptide of the assigned sequence with isotope label measured on the same device for a comparator to and matching the respective spectra, or even a spike-in to the sample.

Unless these doubts can be dispelled somehow, I would have my reservations against the findings presented as mutated HLA ligands here so far.

Minor issues:

Title:

1.) In my view it might make sense to mention that this work is focused on colorectal cancer

Abstract:

2.) It does not make sense to mention 44,785 HLAp – unless the assessed sample numbers are known. It should rather be defined how many HLAp were characterized in one sample, described by a measure of central tendency and variability (e.g. median + IQR, mean + SD).

3.) The authors should introduce and make clear they have assessed HLA class I ligands (by W6/32 antibody precipitation) and this should be defined and explained early on.

Introduction:

4.) What precisely does “for the first time, neoantigens became detectable directly from colorectal cancer (CRC) tissue samples” (line 90-91) actually mean?

Indeed there is no previous report about the identification of mutated neoantigens directly from patients’ samples. However, such mutated neoantigen have been characterized already in organoids from colorectal cancers obtained from patients (i.e. from colorectal tissues) by Alice Newey from the Bassani-Sternberg group (cf. Newey et al. J Immunother Cancer. 2019;7(1):309.). Maybe consider rephrasing to clearly bring your point across.

5.) Consider shortening and revising the introduction structured for main concepts, if possible and deemed useful.

Results:

6.) p. 13 line 175 – 177 is pure speculation. Either explicitly mention this is speculative or omit this, please.

7.) Since this is an essential selling point for your technique: Can the authors provide exact sample weights for the clinical samples (cf. Table 2)

8.) Line 233-335: Maybe move this to the discussion.

Methods

9.) What does “cryopreserved at least once” mean?

10.) Line 397: Has the ethics vote from 2010 been periodically renewed? Is it still valid?

11.) Please define, which mutations have been used to construct personalized databases (small variants e.g. substitutions, indels, frame-shifts or even large rearrangements, fusions etc.)?

12.) Please mention the device and the sequencing approach used for whole exome sequencing also in the main manuscript.

General issues:

13.) I wish to apologize for this in advance, but I cannot get around or neglect the fact that I was not

able to understand several paragraphs in the manuscript and have struggled on several occasions due to issues with English formulations. I believe the authors have done their best but in my view the manuscript would benefit from a thorough proofread e.g. by a native speaker with expertise. This is actually not only about small issues and typos and impairs understandability.

14.) To assume all neoepitopes carry a mutated sequence is meanwhile too simplistic. The authors need to define that they are talking about mutated neoepitopes and use such terms consistently (cf. Finn et al. Cold Spring Harb Perspect Biol. 2018;10(11):a028829. DOI: 10.1101/cshperspect.a028829).

15.) The authors use the term "global-immunopeptidomics" to contrast "targeted-immunopeptidomics". I get the concept of broad vs. targeted mass spectrometry, however global-immunopeptidomics does not seem to work. Rather an untargeted/ shotgun MS approach or even comprehensive MS approach vs. targeted MS seems to make more sense as a concept.

16.) The use of the word "pathological" should be reconsidered, since this is often used conceptually wrong. The authors often rather refer to matched controls (tissue of origin/ malignancy). (e.g. see line 68, line 192)

Side note: How did the author's actually deal with the issue that they had a hepatic metastasis from colorectal cancer. Was the liver tissue the "matched normal" tissue or was colonic tissue available in this case? (cf. line 172)

Marginal points:

17.) Epitopes are usually used for HLA-ligands that are indeed recognized by T cells. Please consider using the term differently (cf. line 181).

18.) Please cite SYFPEITHI (as Rammensee et al. Immunogenetics. 1999;50(3-4):213-9. doi: 10.1007/s002510050595.)

19.) Line 70: Do the authors mean "indispensable" here?

20.) Fig. 3a (Table) Many typos: adebocarcinoma; Net-MHCpan-assined etc.

Reviewer #3 (Remarks to the Author):

The manuscript by Minegishi et al. introduces an optimized method for mass spectrometry based immunopeptidomics analysis using high-field asymmetric waveform ion mobility spectrometry (FAIMS) to perform differential ion mobility (DIM)-MS by seamless gas-phase fractionation. This method enabled the identification of 44,785 HLA-bound peptides (HLAp) from 17 colorectal cancer samples including two neoantigens and revealed cancer-specific processing of HLAp. Interestingly, they further coupled this global-immunopeptidomics analysis with a targeted-immunopeptidomics analysis that is based on screening-oriented reaction monitoring (PRM) to selectively identify particular HLAp of interest. With regard to MS-based analysis of the immunopeptidome of clinical samples, the scarcity of tumor tissue is currently a limiting factor. Therefore, increasing the number of identified HLAp as well as the number of neoantigens is of interest and benefit for the field. Nevertheless, there are many aspects in form and content to be further elucidated. Moreover, a large number of phrases are unclear and/or overstated.

Main manuscript:

- The novelty of the paper is the establishment of a more efficient method for the direct identification of immunopeptides. Therefore, it would be important to explain more in detail the advantages of differential ion mobility and the FAIMS interface and how they work compared to regular MS (L86)
- How does the FAIMS-condition affect the purity and eventually the number of identified peptides predicted as non-binders to defined MHC allotypes in comparison to standard MS? Did the authors perform any experimental comparisons?
- Direct identification of neoantigens from solid tumors has been previously described in an increasing number of publications (L64 => e.g. Ref. 7 and 40). Similarly, neoantigen identification from colorectal cancer samples has been previously published (L90 => doi.org/10.1172/jci.insight.146356.). Please carefully check the literature and put your results in the correct context with

respect to sensitivity of previously published approaches (see also line 154). Similarly, L233-239 represents an overstatement and does not fit to the result chapter but should eventually be part of the discussion.

- Is healthy tissue available for any of the 17 tumor samples? How do the samples correlate per patient (weight)? Please explain Figure 2f and supplementary Figure 3a-e more in detail. How many samples have been included in the analysis? In this regard, please clarify also if paired or unpaired student's t-test has been used (L484)? Supplementary Figure 3b/c: It is kind of clear that the neoantigen has been identified only in tumor samples.
- Information about HLA expression levels on tumor and healthy tissue would be helpful.
- How does the analysis of the immunopeptidome of normal cells from the same patient help to predict the efficacy of ICI in patients? This phrase needs to be described more clearly, also a relevant reference is missing (L52-55).
- Table 1 contains some neoantigens with low affinity: 980.7 nM, 963.6 nM, 728.67 nM. Please comment on these results.
- Concrete affinity values need to be indicated in order to make comparisons understandable (L179).
- L357-365: These are not strong arguments that the peptides can be bound by both HLA allotypes. Affinity prediction and motifs should be provided. Moreover, experimental affinity measurements would be helpful to clearly support the statement.
- More information about spectra of neoantigens identified in tumor samples compared to synthetic peptides should be provided (Supplementary Figure 6 and 7).
- Figure 1 b) and c) number of cells may be indicated more clearly
- Figure 2 d)-f): part of a new chapter. Please consider to change to a new Figure.
- L380-386: Overstatement
- L449: Here it says NetMHCpan4.0, in L.357 it says NetMHCpan 4.1 was used. Please clarify.
- The manuscript requires intensive revision for language and grammar. Please find just selected examples listed below:
 - o L25: these hindrances instead of that hindrances
 - o L58 is the incapability instead of is incapability
 - o L68 establishing a more efficient approach instead of the more efficient approach
 - o L28 Use of correct time: enabled instead of enables
 - o L60: prediction of HLAp in-silico instead of by in-silico
 - o L61: immunopeptides instead of immunopeptidome
 - o L70: indispensable instead of dispensable
 - o L77: mutations rarely exist instead of mutations are rarely exist
 - o L90: ensured a sufficiently deep immunopeptidomics analysis instead of deep immunopeptidome
 - o L121: incomprehensive sentence
 - o L153 compared instead of compare
 - o L158 number of immunopeptides instead of immunopeptide
 - o L159: compared to normal tissue instead of compare to number of normal tissue
 - o L186: increase instead of obtain
 - o L187-190: incomprehensive sentence
 - o L202-207: sentence too long and incomprehensive
 - o L210: compared instead of compare
 - o L221: allotype instead of allotypes
 - o L231: an unknown mechanism instead of the unknown mechanism
 - o L132: In cancer cells instead under cancer environment
 - o L278 and L281: "this result/these results indicated" is used repetitively
 - o L284-287: sentence incomprehensive
 - o L290: "the robust immunopeptidome" what is this?
 - o L302: ensure does not fit in that context
 - o L332: reported instead of reportedly
 - o L337-341: please rephrase
 - o L357: NetMHCpan4.1 instead of NeMHCpan4.1
 - o L484 and L486: Sentence is used repetitively

Supplementary information:

P3 Chapter Differential ion mobility, first sentence: too long and incomprehensive; no mention of a solvent A; what means "to 28"?; how are the results from the three single runs with the three CV sets compiled?

Supplementary Figure 1 and 4: how many peptides were not assigned?

Reviewer #1 (Remarks to the Author):

Review manuscript COMMSBIO-21-3009 T

Minegishi and colleagues describe a FAIMS assisted immunopeptidomics discovery workflow that yields increased numbers of peptides compared to a no FAIMS approach. After evaluating data acquisition strategies on an HCT-116 sample, the authors then employ FAIMS acquisition on a panel of 17 CRC tissues with matched normal. They identify over 40,000 peptides including 2 neoantigens and detect tumor specific antigen processing. They furthermore develop a targeted PRM assay for KRAS derived neoantigens and validate the feasibility in a set of colorectal cancer cell lines. This study employs an innovative data acquisition approach and is of general interest to the field of immunopeptidomics. However, the authors should address the following concerns in a revised version before I can recommend publication of the manuscript.

Major concerns:

- The authors describe a relatively high yield of >3,000 peptides from as little as 5×10^6 cells. For Fig 1b and following, can the authors clarify the input amount and combination of CVs used to achieve this number. It is not clear from the method section whether an equivalent of 5×10^6 cells was injected 3 times for 9 CVs or whether this number is derived by adding up 3 injections of 5×10^6 each to a total of 1.5×10^7 cells. Is the no FAIMS comparison also performed using 3 injections?

➤ Thank you for the comment. In order to clarify the experimental conditions of Figure 1b, we newly prepared the schematic diagram for validation study

which includes the information of injection times and on-column volume (number of cells) that reflects the cell number used per injection as **Supplementary Figure 2a**. We also added the table of peptide numbers in each step of processing to answer your question and for better understanding in **Supplementary Figure 2a**. With FAIMS, on average, more than 3,000 peptides from as little as 5e6 cells (1 ul on-column volume) were identified from each CV Sets. While the 3 replicates without FAIMS with same on-column volume (1 ul) below 3,000 on average. We also modified the main text to mention about this **Supplementary Figure 2a** in the revised version of manuscript of line 124-126.

- What is the overlap of peptides across 9 CVs and which CVs yield the most peptides? What was the cycle time or TopN method used per CV experiment during data acquisition? The vendor recommends 1 additional CV per additional hour of measurement time. Here, 3 CVs are used during a 1 hour gradient. Could one reduce analysis time and increase yield by reducing the CVs sets and the number of injections? Particularly for low input samples, could the yield be increased further by selecting a longer gradient/fewer CVs. Have the authors compared their FAIMS settings with previously published methods for immunopeptidome analysis using FAIMS.

- Thank you for the comment. From our pilot studies to optimize the FAIMS for immunopeptidomics, we checked and optimized the parameters to get the most out of FAIMS-pro interface for immunopeptidomics in advance. We first roughly checked the CV distribution of immunopeptides by PSM counts to choose the best CVs to use. Although the best CV for immunopeptides seemed very variable from peptide to peptide, the substantial PSMs for immunopeptides can be found from CV -80 to CV -35. We rarely found

immunopeptide in CV -90v. Among these, from CV -45v to CV -55v seemed to contain more PSMs for immunopeptides of HCT116. This was described in newly prepared **Supplementary Figure 1a** and we modified the main text accordingly in line 116-117.

This distribution pattern was reproducible in HCT116 and to show the overlap of immunopeptides across 9 CVs, we prepared a new **Supplementary Figure 3c** (HCT116). This was described in revised manuscript, line 148-149. Still, we also noticed this CV preference can be slightly back and forth across samples presumably affected by the different set of HLA allotypes. This difference is described in newly prepared **Supplementary Figure 11c** and **Supplementary Figure 13c** that show the CV distribution in each cell lines. Unlike the HCT116, which prefers CV -50v the best for its immunopeptides, Colo668 and RCM prefer CV -45v the best for their immunopeptides.

- About the cycle time, we followed the publicly available information about multi-CV applied FAIMS+ proteomic analysis, we set the cycle times to 1 sec. (Figure 5a of <https://assets.thermofisher.com/TFS-Assets/CMD/posters/po-65276-faims-orbitrap-fusion-lumos-ms-asms2018-po65276-en.pdf>). In fact, we also tried cycle time 0.75 sec, 1 sec and 2 sec to compare in the optimization step, and we confirmed that the identification number of immunopeptide was most abundant by the cycle time 1 sec. Therefore, we concluded the cycle time 1 sec also works fine with immunopeptidomics analysis. We used 1 hour gradient to make up for the one weakness of FAIMS installation, an attenuated intensity (also shown in Figure 5a of <https://assets.thermofisher.com/TFS-Assets/CMD/posters/po-65276-faims-orbitrap-fusion-lumos-ms-asms2018-po65276-en.pdf>). Since this attenuation is not that severe, this can be easily compensated by increased on-column

volume or shorter gradient. Since our priority in this study is in the immunopeptidomics analysis from the scarce samples, therefore, to keep the intensity, we used 1 hour gradient under FAIMS installation. According to your comment, we added the information of cycle time in **Supplementary Methods, page 2**.

- About the CV combination, we also compared how many CVs are accommodable per injection and checked the best combination that yields the most peptides in advance. According to your comment, we newly added this example information in **Supplementary Figure 1b** and modified the main text accordingly as line 112-113.
- About the comparison of FAIMS settings with previously published methods, we haven't compared the methods so far. There are following reasons we have not compared the methods. First of all, when we started to use FAIMS-pro interface, it was a new device and there was no published method for immunopeptidomics analysis to compare. Second, there is a difficulty in methodological comparison because of the different analytical settings. The best CV settings, CV combinations, cycle time, collision mode and the detector type can be largely different from the lab to the lab due to the type of mass spectrometers. 3rd, most of the previous publications used appreciable amount of sample for immunopeptidomics analyses. And that is not our intention to accomplish in this paper. There is no wonder that if we use more sample amount, more identification can be done. While from the view of clinical practice, it is not always available to secure the huge mass of tumor tissues. We focused especially on this issue to be solved. In addition, most of previous reports used only single CV or at most 2 CVs per run. That may underrepresent the ability of FAIMS device. Therefore, we have not conducted a comparative study so far.

- The authors have uploaded the data to the Japan Proteome Standard Repository/Database (jPOST), however no reviewer login details were provided. Therefore, the quality of the raw data and claims made in the publication could not be evaluated and confirmed. Please provide the raw data to reviewers before publication.

➤ Thank you for the comment. Our apologies for not providing the access keys to the raw files in advance. This was because due to the collaboration request, we are very careful to the accessibility of our raw files before peer-reviewed journal publication. According to your request, we prepared access keys for the jPOST's preview request. Here, you can download the raw data as follows. We would appreciate that the access to the raw file before journal acceptance is only for the review purpose and please refrain from disclosing the information to a third party. We appreciate your understanding.

jPOST Preview Request URLs & Access Keys

ID		jPOST Preview Request	
jPOST-ID	JPST001072	URL:	https://repository.jpostdb.org/preview/42020399625cb5fd503c8
PXD	PXD023770	Access Key	2051
jPOST-ID	JPST001066	URL:	https://repository.jpostdb.org/preview/479044361625cb681beee8
PXD	PXD023771	Access Key	9419
jPOST-ID	JPST001068	URL:	https://repository.jpostdb.org/preview/1568674950625cb6051fa4b
PXD	PXD023773	Access Key	2240
jPOST-ID	JPST001070	URL:	https://repository.jpostdb.org/preview/1427705836625cb5ff2e63e
PXD	PXD023804	Access Key	5126
jPOST-ID	JPST001069	URL:	https://repository.jpostdb.org/preview/143217329625cb6020bf33
PXD	PXD023805	Access Key	9893

- Minegishi et al observe a cancer specific tryptophan trimming of peptides. The trend is already very small, could this not just be due to comparing 5,603 normal exclusive to 14,052 tumor specific peptides? Is this trend also observable when including the shared peptides between tumor and normal tissue?

➤ Thank you for the comment. In connection with the comment by you and the other reviewer, we reanalyzed the data first by comparing the normal and tumor immunopeptidome under shared peptides included condition by paired Student's t test. The new results were shown in the revised version of **Figure 3**. By paired Student's t test, tryptophan showed a tendency of increase in tumor immunopeptidome (**Figure 3f**). Further breakdown of immunopeptides by exclusivity revealed the significantly increased trimming by tryptophan in tumor-exclusive population (**Figure 3g**). Though the statistical difference or the trends are slight as you mentioned, since it was the ratio within independent subpopulations, it can't be denied that these data reflected the unique profile of immunopeptides. Especially, since we also observed an opposite shift of ratio in arginine usage (which became lower in tumor-exclusive population, **Figure 3e**) regardless of the size of subpopulation, we consider these trends of pΩ trimming arose from the concentrated feature of immunopeptides within subpopulation. Since the number of samples were limited in our study, we were also aware of that it is necessary to increase the number of similar analyses. This necessity was already mentioned in our original manuscript and described in a revised manuscript of line 407-408. Based on these new results, we prepared new **Figure 3** and modified the main text accordingly as line 283-288, in our revised manuscript.

- The authors also observed increased cysteine at the C-terminus of tumor

exclusive peptides. As also mentioned in the text, cysteine has not been observed as anchor residue for HLA-I alleles and other studies analyzing HLA bound peptides by mass spectrometry reported poor recovery of cysteines overall. Taken together, this could also indicate that peptides with a C-terminal cysteine are more likely false positive identifications. Did the authors reduce and alkylate to increase yield of cysteine containing peptides?

- Thank you for the comment. We used iodoacetamide (IAA) at final 0.2 mM concentration into lysis buffer just before lysate preparation. We also used protease inhibitor cocktail (Halt™). The description of both additives was missing from the original main text by mistake and accordingly to your comment, we added this description in the revised manuscript of the **Methods** section, line 483-484.

During pilot studies of sample preparation, we noticed that one publication used IAA for immunopeptidomics (*Chong, C. et al., Mol Cell Proteomics, 2018*). There was no clear explanation of adding IAA into lysis buffer from these authors, while it seemed very reasonable to us to add IAA into lysis buffer to avoid the unfavorable disulfide-bond during cellular lysis. Therefore, we conducted the pilot studies for validating the effect of IAA by ourselves and confirmed slightly better overall identification and more cysteine-including immunopeptides. Based on these results, we decided to add IAA into our immunopeptidomics samples. Afterwards, the publication mentioning about the importance of cysteine protection in immunopeptidomics was officially published last year (Sturm, T., et al. *J Proteome Res, 2021*) while we were preparing this manuscript. According to these authors, to avoid the underrepresentation of cysteine residue carrying immunopeptides, it has been recommended that the cysteine protecting agent like iodoacetamide

should be applied during sample preparation. Based on your comment, we added this perspective in revised manuscript, line 394-397.

- This study also describes the development of a KRAS neoantigen specific PRM assay. How many cells were used for the detection of neoantigens in the cell lines studied? It would be interesting to see the results of a PRM analysis using the CRC tissue samples. Could this increase the number of neoantigens discovered for this tumor?

- Thank you so much for the comment. We added the information of the number of cells used (converted from on-column volume) in related figure legends of **Supplementary Figure 6, Supplementary Figure 12 and Supplementary Figure 14**. As you mentioned, it seems very attractive to search shared neoantigens such as oncogenic-KRAS-carrying neoantigens in tissues samples. While unfortunately, there is no spare tissue samples left for that attempt at this time.

Minor concerns:

- The authors comment that true binding peptides might be missed by current prediction algorithms, however, in the methods they state that their analysis is restricted to peptides that predict as binders. How many peptides would they detect if not filtering for predicted ones?

- Thank you for the comment. Since the other reviewer asked about the “purity” of identified peptides by FAIMS-assisted DIM-MS, we added a few representative counts of; peptide identified (as “Protein Groups”), length in 8 to 15 amino acids (as “8-15 aa in length”), deduplicated for unique peptide sequence only (as “Deduplicated”) and assigned-peptides which subtracted

the predicted no-binders by NetMHCpan prediction from “Deduplicated” (as Confirmed-HLAp). According to your comment, we prepared the **Supplementary Figure 2** that describes the representative number and the ratio of no-binders. And we also prepared new **Supplementary Table 1d** that describes each count of tissue samples. During this process, we also noticed that we used incorrect no-binder list for ID172T and ID260N, therefore, we corrected these errors in our revised manuscript. By these corrections the total number of immunopeptides shown in Figure 2i, and the immunopeptidomes of ID 172T and ID260N (Supplementary 8 and 9) were only slightly changed. There was no appreciable influence on our manuscript by this correction.

- From the method section, it is not apparent how much antibody or beads crosslinked with antibody were used per IP.

- Thank you for the comment. According to your comment, we added the information in our revised manuscript, line 486.

- The authors use a stepped elution approach with 20% and 80% acetonitrile. Are both fractions analyzed on the MS or only the 20% fraction?

- We analyzed only 20% fractions for class I immunopeptides. The purpose of 80% ACN elution was to monitor the sample preparation mainly by the amount of κ -chain for the IP efficiency by Western-blotting and silver staining. According to your comment, we added explanation for 20% and 80% eluates in **Supplementary Methods**.

- Could the use of FAIMS benefit the sensitivity of the PRM assay further? Are the same neoantigens detected within the same CV?

➤ Thank you for the comment. As you mentioned, we also believe that there would be a benefit of FAIMS in PRM. The idea of Targeted-MS with FAIMS for oncogenic KRAS neoantigen from clinical tissues is of great interest and we are fully aware of its importance of establishing the methodology in future. While for that purpose, we need more information of KRAS neoantigens, such as peptide sequences and those preferable CVs. Because the optimal CV and the detectable range for peptide are very unique and are impossible to predict in advance without actual measurements at present. We newly prepared **Supplementary Figure 15** to show the examples of uniqueness of preferable CVs and the actually used CVs in neoantigen identification from samples. The neoantigens identified from samples were detected within the CV range that those corresponding synthetic peptides at least so far used for validation, were also be identified. In addition, during the requested validation for neoantigens by other reviewers, the neoantigen KRAS-G13D, which dotp score was turned out to be 0.90 against the synthetic peptide. According to the developer of Skyline, it has been thought that “> 0.90” (e.g., 0.91 or more) is thought to be a good (correct enough) identification of peptide. (<https://skyline.ms/wiki/home/software/Skyline/events/2015%20Webinars/Webinar%2012/page.view?name=qanda>) Thus, it can't be denied that this G13D-carrying peptide was a false-discovery at this point. Therefore, we'd like to withdraw the KRAS-G13D data about KRAS-G13D from our manuscript. In connection with the other reviewers suggestion to mention our work is mainly on colorectal cancer immunopeptidome, we changed the title of manuscript and fit the contents accordingly. According to your comment and the suggestion by reviewers, we newly prepared **Supplementary Figure 15** and described these perspective in our revised manuscript, line 433-452.

- Please copy edit the manuscript for readability to convey the correct meaning of each sentence. As example, line 70 should probably read indispensable instead of dispensable.

- Thank you for the comments. We asked the professional copy editor, *American Journal Expert (AJE)* for English editing and better readability.

Reviewer #2 (Remarks to the Author):

I have had the pleasure to review the article entitled “Comprehensive-immunopeptidomics analysis reveals presentation of potential neoantigens carrying cancer driver mutations“ by Koji Ueda and his group. This group is world-renowned for their work in this field and among the leaders in this field, which is again proven with this work.

I wish to congratulate the authors to their study and the intent to improve HLA-ligandomics with new approaches to yield more data and better insights through additional peptide identifications. This is important work and should be commended. The amount of >6,000 HLA ligands they are able to characterize in minute amounts of samples is truly impressive. Further the combination of different mass spectrometry approaches, e.g. comprehensive analyses and targeted mass spectrometry is a great feature of this presented work.

Obviously, the central aim of their endeavor is to characterize mutated HLA ligands in small tissue samples, which is potentially relevant for individualized therapy approaches. Further they identify mutations in frequent driver mutations (here KRAS in colorectal cancer) that are repeatedly mutated and shown in

different patients.

Major issues

My one central and only major concern to this manuscript is the spectra of the mutated HLA-ligands as provided in Fig. 2 and the supplements. I would like to see some way of validation the presented fragmentation spectra are actually true. For some examples that may prevent confidence, in spite of the strict false discovery rate set to 0.01 to identify peptides: In Fig. 2 C (left lower graph) I can hardly discern the largest fragment at 500.70 m/z (I think), which remains unassigned. Further there are a variety of other small peaks in mutCPPED1 that are unexplained. The mutKRAS sequence in Fig. 2c is extremely repetitive, since it is constituted from 5 of 10 amino acids that are actually all valine in the peptide. This may be an issue since this low diversity may favor false discoveries. In suppl. Fig. 7 the spectra are very busy as well and the largest peak of GAPDH 463.97 m/z remains unexplained, next to two other unassigned peaks. The same goes for CHMP7, where 615.17 m/z remains without explanation. Furthermore, when comparing mutKRAS peptide with the G13D variant (suppl. Fig 7d & 7e), suggested to be repetitive spectra, we can discern a 357.32 m/z and 357.37 m/z peak respectively. The first mentioned peak being the peak with the highest intensity remains unexplained and although generally comparable, for instance regarding assigned peaks the y9 2+ peak 451.95 m/z is the largest peak in e), whereas we only find a small y9 2+ peak in d) with 451.80 m/z, seems relevantly different. The spectra absolutely might be what the authors claim them to be. Nevertheless, there needs to be some kind of validation for sufficient confidence. E.g. one could think of a synthetic peptide of the assigned sequence with isotope label measured on the same device for a comparator to and matching the respective spectra, or even a spike-in to the sample. Unless these doubts can be

dispelled somehow, I would have my reservations against the findings presented as mutated HLA ligands here so far.

- Thank you so much for the important comments. According to your comment, we compared the MS2 spectra by corresponding synthetic peptides. Unfortunately, since there were no spare tissue samples left at this time, the analysis of spiking the stable isotope labeling peptide with sample was impossible. To make the verification more objective, we introduced Skyline software to compare the correlation of dot-product (dotp) score of MS2 spectra by targeted-MS between the sample and the corresponding synthetic peptide for cell line samples. Not only the similar pattern of MS2 spectrum by DIM-MS, 6 examples out of 7 neoantigens we presented in our original manuscript exhibited more than 0.95 dotp score which means confident enough identification of neoantigens from samples. While the neoantigen with KRAS-G13D, which dotp score was turned out to be 0.90 against the synthetic peptide. According to the developer of Skyline, it has been thought that “> 0.90” (e.g., 0.91 or more) is thought to be a good (correct enough) identification of peptide. (<https://skyline.ms/wiki/home/software/Skyline/events/2015%20Webinars/Webinar%2012/page.view?name=qanda>) Thus, the dotp score 0.90 of G13D-carrying neoantigen is not satisfied that threshold. At least from our experience during these validation studies, if the identification is correct enough, the dotp score 0.95 or higher can be easily obtained from the samples, like the other 6 examples. And it can't be denied that this G13D-carrying peptide was a false-discovery as you pointed out. Under this circumstance, it is difficult for us to mention this identification is “correct” with high confidence, therefore, we'd like to withdraw the KRAS-G13D data about

KRAS-G13D from our manuscript. Based on these results of additional experiments, we newly prepared the figure for the validation of MS2 spectra as **Figure 2k-l**, **Supplementary Figure 5**, **Supplementary Figure 6**, **Supplementary Figure 12** and **Supplementary Figure 14**. And modified the main text to fit accordingly as line 154-161, line 220-222, line 309-315, line 327-331 and line 433-439.

About the gray peaks found in MS2 spectra, we consider these were originated from the back grounds which were mixed in the same MS2 window at the same timing as the ion of peptide identified. In our FAIMS methodology, we do not process samples in advance by typical chemical fractionations. Therefore, MS2 spectra from our sample should be more complexed condition so that these backgrounds were observed especially when the identification of peptide from weak precursors. These backgrounds can be found in other publication, for example, Bear et al., reported the MS2 spectra of over expressed KRAS mutants and the synthetic KRAS peptides in the supplementary figure 5 of their publication on *Nature communications* (*Nature Communications*, volume 12, Article number: 4365, 2021), we see the similar gray irrelevant peaks, including sometimes higher than the annotated (b/y ions) MS2 peaks. Since the annotated peaks for neoantigens turned out to reflect the very close pattern of synthetic peptides, we concluded that these gray peaks are possible backgrounds from irrelevant ions, having no critical impact in identification of peptides. According to your comment, we added this description in figure legends of **Figure 2**, **Supplementary Figure 5**, **6**, **12** and **14**.

As results, except for one neoantigen with KRAS-G13D mutation, identified 6 neoantigens out of seven satisfied the identification accuracy. All these 6 neoantigens were identified by global-immunopeptidomics analyses. And the

depth of immunopeptidome obtained by multiple-CV accommodated DIM-MS (global-immunopeptidomics) is still a great advantage for the analysis from scarce samples. We hope these new additional data will resolve your doubts on the identification of peptides by FAIMS-assisted DIM-MS.

Minor issues:

Title:

1.) In my view it might make sense to mention that this work is focused on colorectal cancer

- Thank you for the suggestion. According to your comment, and since we withdraw the data of G13D-carrying neoantigen which was identified by screening oriented targeted-MS, we revised the title as “Differential ion mobility mass spectrometry in immunopeptidomics analysis reveals the presentation of potential neoantigens carrying cancer driver mutations in colorectal cancer” in title page. And based on this modification, we also modified the main text to focus on the colorectal cancer immunopeptidome to fit the title.

Abstract:

2.) It does not make sense to mention 44,785 HLAp – unless the assessed sample numbers are known. It should rather be defined how many HLAp were characterized in one sample, described by a measure of central tendency and variability (e.g. median + IQR, mean + SD).

- > Thank you for the comment. According to your comment, we added the data of immunopeptide per sample in **Figure 2** with box plots (median \pm IQR). The

number of immunopeptide in normal (4378.5 ± 335.8) and tumor tissues (5463.2 ± 367.2) were described with mean \pm SE in revised manuscript, line 180. During this process, we also noticed that we used incorrect no-binder list for ID172T and ID260N, therefore, we corrected these errors in our revised manuscript. By these corrections the total number of immunopeptides shown in Figure 2i, and the immunopeptidomes of ID 172T and ID260N (Supplementary 8 and 9) were only slightly changed. There was no appreciable influence on our manuscript by this correction.

3.) The authors should introduce and make clear they have assessed HLA class I ligands (by W6/32 antibody precipitation) and this should be defined and explained early on.

> Thank you for the comment. According to your comment, we added the description in our revised manuscript, line 112-113.

Introduction:

4.) What precisely does “for the first time, neoantigens became detectable directly from colorectal cancer (CRC) tissue samples“ (line 90-91) actually mean? Indeed there is no previous report about the identification of mutated neoantigens directly from patients’ samples. However, such mutated neoantigen have been characterized already in organoids from colorectal cancers obtained from patients (i.e. from colorectal tissues) by Alice Newey from the Bassani-Sternberg group (cf. Newey et al. J Immunother Cancer. 2019;7(1):309.). Maybe consider rephrasing to clearly bring your point across.

➤ Thank you for the comment. According to your comment and based on the notion, we rephrased the sentence by replacing the words “for the first time” to “from approximately 40 mg of” in line 95-97. From the view of more clinical samples, it is not always available to secure the huge mass of tumor tissues or to establish the alternative patient-derived cultures. We focused especially on this issue to be solved. It has been required to identify neoantigens directly (not bypassing the expansion culture like organoids and xenograft models) from clinical specimens, like microscopic biopsies and the needle biopsies from patients. In connection with your comment, we added this notion in our revised manuscript, line 72-84.

5.) Consider shortening and revising the introduction structured for main concepts, if possible and deemed useful.

➤ Thank you for the comment. According to your comment, we focused on the advantage of DIMS-MS immunopeptidomics to identify neoantigens from scarce samples. On the other hand, one of the reviewers asked us that we should describe more in detail of the meanings of DIM-MS with FAIMS in immunopeptidomics analysis. Therefore, unfortunately, we could not reduce the word counts in the section of Introduction in our revised manuscript.

Results:

6.) p. 13 line 175 – 177 is pure speculation. Either explicitly mention this is speculative or omit this, please.

➤ Thank you for the comment. According to your direction, we omitted the sentence in our revised manuscript.

7.) Since this is an essential selling point for your technique: Can the authors provide exact sample weights for the clinical samples (cf. Table 2)

➤ Thank you for the comments. According to your comments, we added the actual tissues weight of each sample in a new pane as **Supplementary table 1c**. And we added the description in our revised manuscript, line 196-197. And the comparison of normal and tumor tissue weight was depicted as box plots in **Figure 2a**. And we added the description in our revised manuscript, line 177-181.

8.) Line 233-335: Maybe move this to the discussion.

➤ Thank you for the comment. According to you comment, we moved line 235-329 in original manuscript to the discussion section, line 347-353.

Methods

9.) What does “cryopreserved at least once“ mean?

→ Thank you for the comment. Our apologies for unclear writing. We just wanted to say that all the tissues were not immediately processed for immunoprecipitation after the surgical dissection. According to your comment, we revised the sentence to “cryopreserved until use” in our revised manuscript, line 466.

10.) Line 397: Has the ethics vote from 2010 been periodically renewed? Is it still valid?

→ Thank you for the comment. It is still valid and is periodically renewed. According to your comment, we revised the sentence to “This study was first approved by the ethical committee of the Japanese Foundation for Cancer

Research (JFCR) (Ethical committee number 2010-1058) and periodically renewed.” in our revised manuscript, line 474-476

11.) Please define, which mutations have been used to construct personalized databases (small variants e.g. substitutions, inDels, frame-shifts or even large rearrangements, fusions etc.)?

→ Thank you for the comment. In this study, we only used the mutation of amino acid substitution and that was written in the previous version of Supplementary Methods, in the section of “HLA typing, whole exome sequencing and pipe lines for tailored protein database”. While in connection with your next comment on the whole exome sequencing method, we moved the description into our revised manuscript, line 513-515, as well as in line 122-124.

12.) Please mention the device and the sequencing approach used for whole exome sequencing also in the main manuscript.

→ Thank you for the comment. According to your comment, we added the description of whole exome sequencing in our revised manuscript, line 504-518.

General issues:

13.) I wish to apologize for this in advance, but I cannot get around or neglect the fact that I was not able to understand several paragraphs in the manuscript and have struggled on several occasions due to issues with English formulations. I believe the authors have done their best but in my view the manuscript would benefit from a thorough proofread e.g. by a native speaker with expertise. This is actually not only about small issues and typos and impairs understandability.

→ Thank you for the comments. We asked the professional copy editor, *American Journal Expert (AJE)* for English editing and better readability.

14.) To assume all neoepitopes carry a mutated sequence is meanwhile too simplistic. The authors need to define that they are talking about mutated neoepitopes and use such terms consistently (cf. Finn et al. Cold Spring Harb Perspect Biol. 2018;10(11):a028829. DOI: 10.1101/cshperspect.a028829).

➤ Thank you so much for the comment. As you suggested, we introduced more possible sources of neoantigens and the reason why we focused on the neoantigens especially with mutated oncogenic mutations in our revised manuscript, line 47-53 with reference you introduced.

15.) The authors use the term “global-immunopeptidomics” to contrast “targeted-immunopeptidomics”. I get the concept of broad vs. targeted mass spectrometry, however global-immunopeptidomics does not seem to work. Rather an untargeted/ shotgun MS approach or even comprehensive MS approach vs. targeted MS seems to make more sense as a concept.

> Thank you for the suggestion. Since we withdraw the data obtained by screening-oriented targeted-MS, the concept is now on the global-immunopeptidomics DIM-MS analysis in our revised version of manuscript. We'd like to emphasize the importance of choosing immunopeptidomics method appropriately according to the one's purposes. Some researchers need more numbers of variations of immunopeptides (global-immunopeptidomics), for signature/profile analyses. And some are, on the contrary, interested only in the identification of specific immunopeptides of interests, like shared neoantigens (targeted-immunopeptidomics). Global-immunopeptidomics by DIM-MS has its advantage largely in the former purpose and the targeted-immunopeptidomics has, on the contrary, its advantages in the latter purpose. According to your

comment, we added this notion in our revised manuscript, line 347-353 and line 49-452.

16.) The use of the word “pathological“ should be reconsidered, since this is often used conceptually wrong. The authors often rather refer to matched controls (tissue of origin/ malignancy). (e.g. see line 68, line 192) Side note: How did the author’s actually deal with the issue that they had a hepatic metastasis from colorectal cancer. Was the liver tissue the “matched normal“ tissue or was colonic tissue available in this case? (cf. line 172)

> Thank you so much for the sensitive comment. According to your comment, we omitted the word “pathological” from manuscript except for the description of sample state in a Methods section. And in some part, we replaced the “pathological” into more clear description in our revised manuscript as; “normal and tumor” in line 60, “pathological” to “normal” in line 85, “pathological specific” to “cancer-specific” in line 106, “Comparison of pathologically-exclusive immunopeptides revealed the cancer-specific profile of tryptophan trimming at pΩ” to “Comparison of immunopeptides from tissues revealed the cancer-specific profiles of peptide trimming at pΩ” line 248.

We used normal colon tissue for the liver metastasized tumor tissues for sample ID259 & ID261. Since the cell mass of metastasized colon cancer cells originates from the colon epithelial cells, therefore we are in the view of comparison should be done with normal colon tissue that include normal colon epithelial cells.

Marginal points:

17.) Epitopes are usually used for HLA-ligands that are indeed recognized by T cells. Please consider using the term differently (cf. line 181).

→ Thank you for the comment, we understood the term of “epitope” and revised the sentences that included “epitope” by rephrasing or omitting it as follows; “epitope” in original manuscript, line 181 to “peptide” in our revised manuscript, line 233, “in the epitope” in original manuscript, line 186 was omitted in our revised manuscript, line 238.

18.) Please cite SYFPEITHI (as Rammensee et al. Immunogenetics. 1999;50(3-4):213-9. doi: 10.1007/s002510050595.)

→ Thank you for the comment. According to your comment, we added the publication as reference 23 and cited in our revised manuscript, line 230, 556 and 689.

19.) Line 70: Do the authors mean “indispensable” here?

→ Thank you for the comment. Yes and according to your comment, we revised this sentence. We asked the professional copy editor, *American Journal Expert (AJE)* for English editing and better readability.

20.) Fig. 3a (Table) Many typos: adobecarcinoma; Net-MHCpan-assined etc.

→ Thank you for the comment. Our apologies for these typos in our original manuscript. We checked and corrected the typos in our revised version of manuscript.

Reviewer #3 (Remarks to the Author):

The manuscript by Minegishi et al. introduces an optimized method for mass spectrometry based immunopeptidomics analysis using high-field asymmetric waveform ion mobility spectrometry (FAIMS) to perform differential ion mobility

(DIM)-MS by seamless gas-phase fractionation.

This method enabled the identification of 44,785 HLA-bound peptides (HLAp) from 17 colorectal cancer samples including two neoantigens and revealed cancer-specific processing of HLAp. Interestingly, they further coupled this global-immunopeptidomics analysis with a targeted-immunopeptidomics analysis that is based on screening-oriented reaction monitoring (PRM) to selectively identify particular HLAp of interest. With regard to MS-based analysis of the immunopeptidome of clinical samples, the scarcity of tumor tissue is currently a limiting factor. Therefore, increasing the number of identified HLAp as well as the number of neoantigens is of interest and benefit for the field.

Nevertheless, there are many aspects in form and content to be further elucidated. Moreover, a large number of phrases are unclear and/or overstated.

Main manuscript:

- The novelty of the paper is the establishment of a more efficient method for the direct identification of immunopeptides. Therefore, it would be important to explain more in detail the advantages of differential ion mobility and the FAIMS interface and how they work compared to regular MS (L86)

→ Thank you for the comment. According to your comment, we modified the paragraph of the original manuscript, line 86 to pointing out the current task raised in the field of immunopeptide by international consortium of immunopeptidomics in our revised manuscript, line 91-101. We also added the background of immunopeptidomics to better understand why FAIMS has advantages in our revised manuscript, line 72-84.

- How does the FAIMS-condition affect the purity and eventually the number of identified peptides predicted as non-binders to defined MHC allotypes in

comparison to standard MS? Did the authors perform any experimental comparisons?

→ Thank you for the insightful comment. We performed the experimental comparisons for the first step which is shown in **Figure 1b to 1e**. We performed with or without FAIMS condition from the same sample and the same injection volume. The experimental details were newly added as **Supplementary Figure 2a** and the number of peptides at each processing steps were also exhibited in table format for better understanding. While we didn't pay so much attention to the "purity" of identified peptides under FAIMS setting, according to your comment, we compared the proportion of assigned peptides against the "Peptide Group" and the "Deduplicate" stages with or without FAIMS condition. And unexpectedly, it turned out not just increasing the number of peptides identified ("Peptide Groups"), but also the FAIMS positively affected the "purity" of peptide identified, i.e., included less no-binders (shown in **Supplementary Figure 2b and 2c**) when comparing the analyses of with or without FAIMS (n= 9 each). Though the difference was very small (+0.48 % better in purity with FAIMS condition by "No-binder to Peptide Groups" ratio and + 0.45 % better in purity with FAIMS condition by "No-binder to Deduplicate" ratio), this awareness is very important because we speculate that one of the benefits of FAIMS in immunopeptidomics was provably shown in this difference. According to the vendor of FAIMS-pro interface, it has been reported that "FAIMS produces superior peak purity for low abundance precursor" (<https://assets.thermofisher.com/TFS-Assets/CMD/posters/po-65276-faims-orbitrap-fusion-lumos-ms-asms2018-po65276-en.pdf>, Figure 5B). This feature appears exactly to be suitable for the immunopeptides from samples with limited amounts. Based on these results, in addition to newly prepared **Supplementary**

Figure 2b and 2c, we added the description for FAIMS effect in our revised manuscript, line 138-145.

- Direct identification of neoantigens from solid tumors has been previously described in an increasing number of publications (L64 => e.g. Ref. 7 and 40). Similarly, neoantigen identification from colorectal cancer samples has been previously published (L90 => doi.org/10.1172/jci.insight.146356.). Please carefully check the literature and put your results in the correct context with respect to sensitivity of previously published approaches (see also line 154). Similarly, L233-239 represents an overstatement and does not fit to the result chapter but should eventually be part of the discussion.

7→8, 40→ 16

> Thank you so much for the comments. In Ref. 7 (now Ref. 8 in our revised manuscript), neoantigen identification from solid tumor of melanoma samples and that was why we excluded the melanoma from mentioning as unsuccessful previous examples in our main text. In their methods, the authors mentioned that 100-4000 mg of tissue was necessary for their immunopeptidomics analyses. We couldn't find the exact weight used for neoantigen-identified samples, still the smallest amount (100 mg) they used was larger than we used. In Ref 40 (now Ref. 19 in our revised manuscript), the authors identified neoantigen from patient derived primary cancer cell lines, not directly from tissue samples. Patient-derived primary cancer cell lines, organoids and the xenograft models are the important technique to secure the clinical materials. Still, the cancer microenvironment which has impacts on immunopeptide presentation in tumor tissue, will be lost. We believe getting a whole picture of immunopeptidome at tissue level is critical for future study to extract the immunopeptidomics profile which associates with ICB response or severe adverse events. About the JCI

report, the authors mentioned that they used more than 1.5 g of tissues for their immunopeptidomics analyses. Since we used 42.9 mg of tissue on average in our study, these are more than 35 times larger amounts of material sources. The 40 mg of tissue is a biopsy-friendly size and the immunopeptidomics analysis which has sensitivity enough to identify neoantigen directly, this means more opportunity for patients with who suffer small size of tumors. We believe our study is unique in this notion and of interest to researchers who are interested in immunopeptidomics analyses from limited sample amount. According to your comment, we revised paragraph by comparing the past method in our revised manuscript line 72-86. We also added the publication of JCI (doi.org/10.1172/jci.insight.146356.) into our manuscript as reference 16 and sited in our revised manuscript, line 76. And we moved line 233-239 in the section of discussion in our revised manuscript, line 247-353, accordingly to your comment. One excuse if we may add, when we first posted the early version of this manuscript in bioRxiv (April, 2021), the publication by *Hirama et al* in JCI insight was still not open to public. Our apologies for not referred this literature in our original manuscript.

- Is healthy tissue available for any of the 17 tumor samples?
 - Thank you for the comment. According to your comment, we added the description “Among obtained clinical tissue sample sets, 15 out of 17 were the set of primary colon tumor tissue with normal colon tissue. The other 2 out of 17 were the set of liver-metastasized colon tumor tissue with normal colon tissues. According to your comment, we added this information in the main text of **Methods** section, line 467-470.

- How do the samples correlate per patient (weight)?

➤ Thank you for the comment. According to your comment, we assessed the correlation between the number of immunopeptide identified and the weight of tissue used. As shown in newly added **Supplementary Figure 7c**, there was no statistically significant correlation ($r = -0.121$, $p = 0.496$) between the number of immunopeptides identified and the tissue weight. We added this results in the main text line 195-196.

• Please explain Figure 2f and supplementary Figure 3a-e more in detail.

➤ Thank you for the comment. Associating with other reviewer's comment, we moved former Supplementary Figure 3a-e as main **Figure 2a-d** with additional data and added the description in our revised main text, line 177 - 199. We also added more explanation for former Figure 2f (now **Figure 3h** in our revised manuscript), line 288-293.

• How many samples have been included in the analysis? In this regard, please clarify also if paired or unpaired student's t-test has been used (L484)?
Supplementary Figure 3b/c: It is kind of clear that the neoantigen has been identified only in tumor samples.

➤ Thank you for your comment. We used unpaired t-test in the original version of our manuscript. While you made us notice that this shift of trimming at $p\Omega$ should have been done by paired Student's t test, we first reanalyzed the data between normal immunopeptidome and tumor immunopeptidome per individual by paired Student's t test. The new results were prepared as **Figure 3**. The sample that contained at least one or more amino acids of interest at $p\Omega$ was included in the analyses. We added this information in our revised manuscript, line 261 for $p\Omega$ -Cys, line 274 for $p\Omega$ -Arg and line 284 for $p\Omega$ -Trp. Based on these results, we then classified immunopeptide by exclusivity to

reveal where the difference originates from and this was also analyzed by paired Student's t-test. We added this description in a section of revised **Methods**, line 591-592.

- Information about HLA expression levels on tumor and healthy tissue would be helpful.

➤ Thank you so much for the comment. Based on your comment, we conducted Western blotting against α -chain of HLA complex to assess the overall picture of HLA amount. To gain the relative quantity (RQ) of α -chain between samples, we loaded the same master sample (lysate of HCT116) in every gel to calculate RQ. All blot images used were shown in newly added **Supplementary Figure 7**. From results, we found that the RQ of α -chain was more abundant in tumor tissues. When we normalized the number of immunopeptide identified in sample by RQ of α -chain, there was no significant difference in identified number of immunopeptides between normal and tumor samples. Further analysis revealed that the number of immunopeptide identified showed stronger positive correlation to the RQ of α -chain ($r = 0.64$, $p = 4.49E-05$) than the protein amount ($r = 0.36$, $p = 0.04$). These results were newly added in our revised manuscript, **Figure 2e to 2h**. And revised the main text to fit the contents accordingly as, line 186-195.

- How does the analysis of the immunopeptidome of normal cells from the same patient help to predict the efficacy of ICI in patients? This phrase needs to be described more clearly, also a relevant reference is missing (L52-55).

➤ Thank you for the comment. According to your comment, we revised the sentence as “From these perspectives, the comparative analysis of normal and tumor tissue-based immunopeptidomes from identical patients is

increasingly important to avoid the adverse events of ICI in patients⁴.” in our revised manuscript, line 59-61. Since the HLA genotype is quite diverse from individual to individual, as well as in the race to race, the proper control of normal immunopeptidome is critical to delineate the possible accurate signatures of ICB responding or adverse events signature. From this perspective, we think it is important to conduct a personal immunopeptidomics analyses from normal and diseased condition. We added this notion in our revised manuscript, line 400-404.

- Table 1 contains some neoantigens with low affinity: 980.7 nM, 963.6 nM, 728.67 nM. Please comment on these results.

➤ Thank you for the comments. Our apologies for the incorrect description of filtering of immunopeptides in our original manuscript. We corrected the filtering description in our revised manuscript, line 547-552. The threshold 500 nM is a recommended threshold in the previous version of NetMHC (NetMHC3.0) by the authors, Lundegaard et al. (Nucleic Acid Res, 2008).

While for the clinical samples, since the genetic background of HLA is quite diverse, it is convenient to use the NetMHCpan to predict binders. Because that covers more variation of HLA allotypes. Therefore, we used NetMHCpan4.1 to predict the binders for the first place in this report. We then used NetMHC4.0 to know the insights of affinity if it's available.

The NetMHC prediction includes not a small number of immunopeptides which have weaker affinity than 500 nM as binders. Therefore, the prediction gaps, like some predicted-binders with lower affinity (> 500 nM), between the NetMHCpan and the NetMHC were occasionally found.

According to your comment, we also added the description of prediction gaps between NetMHC and NetMHCpan in our revised manuscript, line 423-428.

- Concrete affinity values need to be indicated in order to make comparisons understandable (L179).

> Thank you for the comment. According to your comment, we prepared the new comparative table that shows the binding prediction by NetMHCpan4.1 and NetMHC4.0 for all identified neoantigens as well as to the corresponding wild type sequences in this study. The information of affinity prediction you commented was also included if its available by NetMHC4.0. These tables were newly prepared as **Table 1**, **Table 2** and **Table 3**.

- L357-365: These are not strong arguments that the peptides can be bound by both HLA allotypes. Affinity prediction and motifs should be provided. Moreover, experimental affinity measurements would be helpful to clearly support the statement.

➤ Thank you for the comment. In connection with your previous comments, we prepared new tables that includes these information (**Table 1**, **Table 2** and **Table3**). And thank you for the critical comments of affinity measurements about the neoantigens. We are really interested in the experimental validation of affinity, but unfortunately, we could not find an access to a good collaborator at this moment who would be able to accomplish the experimental validation within the revising period for this paper. While according to your comment, we added the notion about the importance of experimental affinity measurements in our revised manuscript, line 428-430

- More information about spectra of neoantigens identified in tumor samples compared to synthetical peptides should be provided (Supplementary Figure 6 and 7).

- Thank you so much for the comments. According to your comment, we compared the MS2 spectra by corresponding synthetic peptides. To make the verification more objective, we introduced Skyline software to compare the correlation of dot-product (dotp) score of MS2 spectra by targeted-MS between the sample and the corresponding synthetic peptide for cell line samples. Not only the similar pattern of MS2 spectrum by DIM-MS, 6 examples out of 7 neoantigens we presented in our original manuscript exhibited more than 0.95 dotp score which means confident enough identification of neoantigens from samples. While the neoantigen with KRAS-G13D, which dotp score was turned out to be 0.90 against the synthetic peptide. According to the developer of Skyline, it has been thought that “> 0.90” (e.g., 0.91 or more) is thought to be a good (correct enough) identification of peptide. (<https://skyline.ms/wiki/home/software/Skyline/events/2015%20Webinars/Webinar%2012/page.view?name=qanda>) Thus, the dotp score 0.90 of G13D-carrying neoantigen is not satisfied that threshold. At least from our experience during these validation studies, if the identification is correct enough, the dotp score 0.95 or higher can be easily obtained from the samples, like the other 6 examples. And it can't be denied that this G13D-carrying peptide was a false-discovery as you pointed out. Under this circumstance, it is difficult for us to mention this identification is “correct” with high confidence, therefore, we'd like to withdraw the KRAS-G13D data about KRAS-G13D from our manuscript. Based on these results of additional experiments, we newly prepared the figure for the validation of MS2 spectra as **Figure 2k-l**, **Supplementary Figure 5**, **Supplementary Figure 6**, **Supplementary Figure 12** and **Supplementary Figure 14**. And modified the

main text to fit accordingly as line 154-161, line 220-222, line 309-315, line 327-331 and line 433-439.

- Figure 1 b) and c) number of cells may be indicated more clearly

➤ Thank you for the comment. In order to clarify the number of cells in Figure 1, we newly prepared the schematic diagram for validation study which includes the information of injection times and on-column volume (number of cells) that reflects the cell number used per injection as **Supplementary Figure 2a**. We also modified the main text to mention about this **Supplementary Figure 2a** in the revised version of manuscript of line 124-126.

- Figure 2 d)-f): part of a new chapter. Please consider to change to a new Figure.

>Thank you for the comment. According to your direction, we separated the former Figure 2d and 2f as a new **Figure 3** in revised version.

- L380-386: Overstatement

➤ Thank you for the comment. According to your comment, we omit the paragraph and revised in our revise manuscript, line 458-461.

- L449: Here it says NetMHCpan4.0, in L.357 it says NetMHCpan 4.1 was used. Please clarify.

➤ Thank you for the comment. It's done by NetMHCpan4.1. According to your comment, we corrected the description in our revised manuscript, line 548.

- The manuscript requires intensive revision for language and grammar. Please find just selected examples listed below:

- o L25: these hindrances instead of that hindrances

- o L58 is the incapability instead of is incapability
- o L68 establishing a more efficient approach instead of the more efficient approach
- o L28 Use of correct time: enabled instead of enables
- o L60: prediction of HLAp in-silico instead of by in-silico
- o L61: immunopeptides instead of immunopeptidome
- o L70: indispensable instead of dispensable
- o L77: mutations rarely exist instead of mutations are rarely exist
- o L90: ensured a sufficiently deep immunopeptidomics analysis instead of deep immunopeptidome
- o L121: incomprehensive sentence
- o L153 compared instead of compare
- o L158 number of immunopeptides instead of immunopeptide
- o L159: compared to normal tissue instead of compare to number of normal tissue
- o L186: increase instead of obtain
- o L187-190: incomprehensive sentence
- o L202-207: sentence too long and incomprehensive
- o L210: compared instead of compare
- o L221: allotype instead of allotypes
- o L231: an unknown mechanism instead of the unknown mechanism
- o L132: In cancer cells instead under cancer environment
- o L278 and L281: "this result/these results indicated" is used repetitively
- o L284-287: sentence incomprehensive
- o L290: "the robust immunopeptidome" what is this?
- o L302: ensure does not fit in that context
- o L332: reported instead of reportedly

- o L337-341: please rephrase
- o L357: NetMHCpan4.1 instead of NeMHCpan4.1
- o L484 and L486: Sentence is used repetitively
- Thank you for the comments. According to your comment, we first corrected the errors you mentioned above. And then, we asked the professional copy editor, *American Journal Expert (AJE)* for English editing and better readability.

Supplementary information:

P3 Chapter Differential ion mobility, first sentence: too long and incomprehensive; no mention of a solvent A; what means “to 28”?; how are the results from the three single runs with the three CV sets compiled?

- Thank you for the comment. According to your comment, we revised the chapter of supplementary method and added the description of solvent A. The description of “to 28” was corrected as “28% solvent B” in **Supplementary Methods**. The results of single runs with the three CV sets were newly added in the **Supplementary Figure 2** as a schematic diagram for FAIMS validation analyses.

➤

Supplementary Figure 1 and 4: how many peptides were not assigned?

- Thank you for the comment. According to your comment, we prepared a new **Supplementary Table 1e** that shows the number of no-binder by NetMHCpan prediction. During this process, we also noticed that we used incorrect no-binder list for ID172T and ID260N, therefore, we corrected these errors in our revised manuscript. By these corrections the total number of immunopeptides shown in Figure 2i, and the immunopeptidomes of ID 172T

and ID260N (Supplementary 8 and 9) were only slightly changed. There was no appreciable influence on our manuscript by this correction.

REVIEWERS' COMMENTS:

Reviewer #2 (Remarks to the Author):

I have reassessed the article submitted by Minegishi et al. with the revised title "Differential ion mobility mass spectrometry in immunopeptidomics analysis reveals the presentation of potential neoantigens carrying cancer driver mutations in colorectal cancer".

The authors have addressed the comprehensive review of their manuscript provided by three experts in the field.

My critique has been adequately addressed and in my personal view the authors have also comprehensively addressed the valid critique and issues raised by the other reviewers, including introducing improvements of the language (with external support) as well as validation efforts of the presented mutated neoepitopes. The former has substantially improved ease of understanding and readability of the manuscript in my view.

Further the authors have now included additional relevant data in the revised supplementary materials.

I am convinced this review process and the diligent work of the authors has relevantly improved the manuscript, which I truly appreciate.

Please do still consider the following small points:

I assume that the requested access to the raw data files currently deposited on jPOST will be provided once the manuscript is published.

Introduction line p. 3, l. 52: a word such e.g. "like" seems to be missing.

Since it is indeed a very relevant challenge to improve the identification of (mutated) neoantigens directly from limited small clinical samples, I thank the authors for their commendable diligence and their work as well as the comprehensive revisions of their manuscript.

I wish the authors the best of success with continuing their relevant work.

Reviewer #3 (Remarks to the Author):

Thank you again for giving me the opportunity to review the revised manuscript by Minegishi et al. In fact, the manuscript improved substantially by the revision and my previous concerns have been addressed in a fully satisfying manner. With respect to language the manuscript also improved significantly. The authors may consider rephrasing the following sentences:

- "or the normal proteins but epigenetically differentially increased proteins" (L48)
- "is required never before" (L52)
- "For the sample preparation of Class I immunopeptide in our study, we used 113 immunopurified the HLA complex by W6/32 antibody" (L112)